# Upscaling dryland carbon and water fluxes with artificial neural networks of optical, thermal, and microwave satellite remote sensing

Matthew P. Dannenberg[1], Mallory L. Barnes[2], William K. Smith[3], Miriam R. Johnston[1], Susan K. Meerdink[1], Xian Wang[2,3], Russell L. Scott[4], Joel A. Biederman[4]

[1]Department of Geographical and Sustainability Sciences, University of Iowa, Iowa City IA 52245, USA
[2]O'Neill School of Public and Environmental Affairs, Indiana University, Bloomington IN 47405, USA
[3]School of Natural Resources and the Environment, University of Arizona, Tucson AZ 85721, USA
[4]Southwest Watershed Research Center, Agricultural Research Service, U.S. Department of Agriculture, Tucson AZ 85719, USA

*Correspondence to*: Matthew P. Dannenberg (matthew-dannenberg@uiowa.edu)

**Abstract.** Earth's drylands are home to more than two billion people, provide key ecosystem services, and exert a large influence on the trends and variability in Earth's carbon cycle. However, modeling dryland carbon and water fluxes with remote sensing suffers from unique challenges not typically encountered in mesic systems, particularly in capturing soil moisture stress. Here, we develop and evaluate an approach for joint modeling of dryland gross primary production (GPP), net ecosystem exchange (NEE), and evapotranspiration (ET) in the western United States (U.S.) using a suite of AmeriFlux eddy covariance sites spanning major functional types and aridity regimes. We use artificial neural networks (ANNs) to predict dryland ecosystem fluxes by fusing optical vegetation indices, multitemporal thermal observations, and microwave soil moisture/temperature retrievals from the Soil Moisture Active Passive (SMAP) sensor. Our new dryland ANN (DrylANNd) carbon and water flux model explains more than 70% of monthly variance in GPP and ET, improving upon existing MODIS GPP and ET estimates at most dryland eddy covariance sites. DrylANNd predictions of NEE were considerably worse than its predictions of GPP and ET, likely because soil and plant respiratory processes are largely invisible to satellite sensors. Optical vegetation indices, particularly the normalized difference vegetation index (NDVI) and near-infrared reflectance of vegetation ($NIR_v$), were generally the most important variables contributing to model skill. However, daytime and nighttime land surface temperatures and SMAP soil moisture and soil temperature also contributed to model skill, with SMAP especially improving model predictions of shrubland, grassland, and savanna fluxes and land surface temperatures improving predictions in evergreen needleleaf forests. Our results show that a combination of optical vegetation indices, thermal infrared, and microwave observations can substantially improve estimates of carbon and water fluxes in drylands, potentially providing the means to better monitor vegetation function and ecosystem services in these important regions that are undergoing rapid hydroclimatic change.

## 1 Introduction

Earth's drylands are critically important to society yet exceptionally vulnerable to climate change. Drylands are home to more than two billion people and make up more than 40% of Earth's land surface (Reynolds et al., 2007). Primary production of dryland vegetation supports many rare and endemic species as well as extensive rangelands and croplands (Bestelmeyer et al., 2015). Dryland ecosystems are also important regulators of global trends and interannual variability of Earth's carbon cycle (Humphrey et al., 2018; Ahlström et al., 2015; Poulter et al., 2014), due both to their large spatial extent and high climate sensitivity (Biederman et al., 2016; Zhang et al., 2022). Hotter and atmospherically drier conditions associated with anthropogenic climate change will likely increase water limitation (Cayan et al., 2010; Cook et al., 2015; Williams et al., 2020; Ault, 2020; Cook et al., 2020), possibly leading to expansion and degradation of drylands (Huang et al., 2016, 2017). There is therefore a pressing need for satellite-based monitoring of dryland carbon and water cycling at large scales.

While many remote sensing techniques were originally developed and tested in drylands (e.g., Huete, 1988; Huete & Jackson, 1987; Rouse et al., 1974), satellite-based modeling of dryland carbon and water fluxes has been a long-standing challenge. For example, early validation studies of the Moderate Resolution Imaging Spectroradiometer (MODIS) science products noted tendencies to overestimate mean dryland productivity (Heinsch et al., 2006; Turner et al., 2005, 2006a, b) and to miss important features of the seasonal cycle (Heinsch et al., 2006; Turner et al., 2006b). Recent work has also shown that both satellite models (Biederman et al., 2017; Stocker et al., 2019) and process-based models (MacBean et al., 2021) dramatically underestimate the interannual variability of dryland carbon and water fluxes, while also frequently failing to capture the "flashy" and multi-modal seasonal dynamics of dryland carbon cycling (Barnes et al., 2021). For instance, the widely used MODIS gross primary production (GPP) and evapotranspiration (ET) products substantially underestimate the variability of carbon and water fluxes of the western US, capturing only ~30% of interannual variability (Biederman et al., 2017).

Several issues make drylands uniquely difficult to monitor and model with remote sensing (Smith et al., 2019). First, ecosystem carbon and water exchange are more tightly coupled to soil moisture in drylands than in wetter, more mesic systems where moisture tends to be more plentiful (Novick et al., 2016; Stocker et al., 2018), but most existing satellite-based models do not explicitly represent soil moisture stress (Song et al., 2013). Instead, light-use efficiency models often represent moisture stress using vapor pressure deficit (Running et al., 2004; Zhang et al., 2016). While vapor pressure deficit is well suited as a water stress indicator for mesic regions, it often does not fully capture water stress in drylands, where soil moisture plays a particularly important role in regulating surface conductance and carbon and water fluxes (Novick et al., 2016; Stocker et al., 2018; Dannenberg et al., 2022). Soil moisture therefore needs to be incorporated into satellite-based carbon and water models to represent temporal variability of dryland water limitation (Stocker et al., 2018, 2019; Smith et al., 2019).

Second, dryland plants have physiological responses to water limitation (and precipitation variability more generally) that are not necessarily captured by standard remote sensing approaches. Many dryland plants have drought adaptations that allow them to remain green even while functionally inactive under extreme moisture stress (Yan et al., 2019; Smith et al., 2019),

making it difficult to resolve temporal variation in dryland plant function. Therefore, plant physiological responses to periods of moisture stress are not necessarily reflected in optical vegetation indices (VIs) (Yan et al., 2019; Wang et al., 2022; Smith et al., 2018). The normalized difference vegetation index (NDVI) is the most widely used vegetation index, but it sometimes fails to capture temporal dynamics of carbon and water fluxes in drylands (Yan et al., 2019; Smith et al., 2019; Wang et al., 2022). While other optical vegetation indices overcome some of the weaknesses of NDVI, combining different types of

remotely sensed observations—such as those from microwave, thermal, and visible wavelengths—can capture complementary information about plant and ecosystem stress that is unattainable from optical VIs alone (Smith et al., 2019; Stavros et al., 2017; Guan et al., 2017). Land surface temperature (LST) from thermal imaging, for example, is an important determinant of carbon and water fluxes because, among other reasons, both photosynthesis and respiration involve temperature-dependent enzymatic reactions (Farquhar et al., 1980; Atkin and Tjoelker, 2003) and because it is a key indicator of latent heat flux,

which cools leaves and land surfaces (Bateni and Entekhabi, 2012). Integration of multi-source satellite remote sensing could therefore improve representation of plant physiological responses to periodic moisture stress in drylands as compared with optical VIs alone.

Third, drylands tend to be more spatially heterogeneous than many other ecosystems, consisting of complex mixtures of

vegetation structural, morphological, functional, and physiological characteristics that vary over relatively short distances. These mixtures of vegetation types within moderate- to coarse-resolution imagery can contribute to significant error in GPP estimates (Turner et al., 2002; Heinsch et al., 2006). Many large-scale remote sensing-based carbon and water models assume a single vegetation type for each coarse pixel rather than representing the land surface as a continuous mixture of different cover types. In the open canopies typical of dryland ecosystems, optical VIs are also particularly sensitive to soil background

reflectance and the presence of senesced vegetation or standing litter (Huete and Jackson, 1987). High spatial heterogeneity in dryland vegetation, in combination with complex terrain in some areas, leads to diverse ecosystem seasonalities. Drylands in the western U.S. often have one or more annual growing seasons occurring in spring and/or summer (Biederman et al., 2017; Dannenberg et al., 2020), and the timing, length, and productivity of those growing seasons can vary substantially from year to year in response to ocean-atmosphere teleconnections (Dannenberg et al., 2015, 2021). Moreover, carbon and water fluxes

in drylands depend on intermittent and highly variable "pulses" of precipitation that are less seasonally and spatially uniform than limiting resources (e.g., temperature and light) in more mesic or temperate ecosystems (Huxman et al., 2004; Roby et al., 2020). This combination of high spatial and temporal heterogeneity in dryland ecosystem structure and function leads to highly "unique" patterns in carbon and water fluxes, meaning that models perform poorly when used to predict fluxes at sites on which they are not trained (Haughton et al., 2018), yet the flux tower networks typically used to train remote sensing-based

models have notably low representation of dryland sites relative to their global prevalence (Smith et al., 2019). The spatial and

temporal heterogeneity of limiting resource availability, "uniqueness" of dryland fluxes to their specific location (i.e., low predictive power of models for sites on which they were not trained), and relatively sparse dryland observation networks combine to increase uncertainty in carbon and water cycling estimates from models primarily calibrated for other regions.

With new sensors, new vegetation indices, and expanded global ground networks, many of these issues are now at least partly addressable. Recent research has focused on using different combinations of remote sensing data, including integration of soil moisture (Stocker et al., 2019; Jones et al., 2017), multispectral (Barnes et al., 2021), and thermal infrared (Sims et al., 2008; Anderson et al., 2012) observations in models ranging in complexity from purely empirical to semi-empirical or process-based. For example, dryland-specific GPP estimates based on machine learning of meteorological reanalysis data and optical remote

sensing observations outperform globally-trained models at capturing seasonal to interannual variability of dryland GPP (Barnes et al., 2021). New satellite microwave missions also allow more direct sensing of soil moisture than previously available (Song et al., 2013; Jones et al., 2017; Smith et al., 2019), which could address one of the biggest contributors to model error in dryland ecosystems: the tight coupling between plant activity and soil moisture that is not well-captured by vapor pressure deficit (Novick et al., 2016; Stocker et al., 2018; Heinsch et al., 2006) or remotely sensed greenness (Yan et al.,

110 2019).

Here, we aim to improve estimation of dryland GPP, net ecosystem exchange (NEE), and evapotranspiration using an extensive network of eddy covariance observations and multi-source satellite remote sensing. We specifically develop and test an approach for data-driven prediction of a full suite of carbon and water fluxes that are specifically adapted for drylands using a

machine learning fusion of multispectral, thermal, and microwave remote sensing. We use an ensemble of artificial neural networks (ANNs) to jointly predict the key ecosystem carbon and water fluxes—GPP, NEE, and ET—at monthly, 0.05° resolution using a combination of optical VIs from MODIS, daytime and nighttime MODIS LST, soil moisture and soil temperature from the Soil Moisture Active Passive (SMAP) sensor, and subpixel fractional land cover. We develop and test the dryland ANN model (hereafter called "DrylANNd") using flux observations from 28 AmeriFlux eddy covariance towers

in arid to subhumid regions of the western United States. We evaluate the ability of the model to capture monthly variability (§3.1), spatial patterns (§3.1), seasonality (§3.2), and interannual variability (§3.3) of GPP, NEE, and ET; compare the model estimates to existing MODIS GPP and ET products (§3.1-3.3); and assess which remotely sensed variables are most important for improving GPP, NEE, and ET estimates in drylands (§3.4). Our data-driven model will provide new and improved estimates of the variability and hydroclimatic drivers of carbon and water fluxes across the western U.S., with the potential to inform

and develop future global-scale carbon and water flux estimates.

## 2 Materials and Methods

### 2.1 Study area and eddy covariance data

We developed and tested DrylANNd across 28 AmeriFlux eddy covariance sites (Fig. 1; Table S1), each overlapping the SMAP record (2015-present) by at least one full year and consisting predominantly of natural vegetation. Based on the 1981-
2010 TerraClimate annual precipitation (P) and potential evapotranspiration (PET) normals (Abatzoglou et al., 2018), most sites are semiarid ($0.2 \leq$ P/PET $< 0.5$; $N$=16), with several others classified as arid ($0.03 \leq$ P/PET $< 0.2$; $N$=6) and subhumid ($0.5 \leq$ P/PET $\leq 0.75$; $N$=5). One site (Valles Caldera Mixed Conifer; US-Vcm) is slightly wetter than subhumid (P/PET $= 0.85$) due to its relatively high elevation (>3,000 m), but we include it here because it is part of the six-site New Mexico Elevation Gradient network (Anderson-Teixeira et al., 2011) and is frequently used in U.S. dryland flux research (e.g., Biederman et al.,
2017, 2016). The sites span a large latitudinal gradient (31.74°N to 46.69°N) and include six evergreen needleleaf forest (ENF), seven grassland (GRS), ten shrubland (SHB), and five savanna (SAV) sites.

We used a spike detection method to filter out sudden but temporary changes in half-hourly NEE, which can arise either from biophysical effects (e.g., sudden changes in turbulence) or from instrument error (Papale et al., 2006). Using REddyProc
(Wutzler et al., 2018, 2020) in the R statistical computing environment (R Core Team, 2021), we then excluded half-hourly NEE observations that occurred during periods of low turbulence based on a seasonal friction velocity (U*) filter, defined empirically based on the U* distribution within each site and season using the moving point method (Wutzler et al., 2018; Papale et al., 2006), and gap-filled the missing data using a look-up table based on air temperature, shortwave radiation, and vapor pressure deficit (Papale et al., 2006). We also gap-filled latent heat flux estimates using the same method and converted
latent heat flux to ET (in mm). We partitioned the half-hourly NEE into its component parts (GPP and ecosystem respiration) using the nighttime partitioning method (Papale et al., 2006; Reichstein et al., 2005), summed the half-hourly GPP, NEE, and ET to total daily fluxes, and calculated mean daily fluxes for each calendar month.

### 2.2 Remote sensing data

For each site, we obtained daily 500-meter multispectral surface reflectance from the MODIS Nadir Bidirectional Reflectance
Distribution Function (BRDF)-Adjusted surface Reflectance (NBAR) product (MCD43A4) (Schaaf et al., 2002), excluding observations for which only fill values were provided (i.e., where a BRDF inversion could not be achieved) or where reflectance was below zero. For regional-scale prediction, we used the 0.05° MODIS Climate Modeling Grid (CMG) version of the NBAR product (MCD43C4). From the seven-band surface reflectance, we calculated seven daily VIs (Table 1): NDVI (Rouse et al., 1974; Tucker, 1979), enhanced vegetation index (EVI) (Huete et al., 2002), near-infrared reflectance of
vegetation (NIR$_v$) (Badgley et al., 2017), kernel NDVI (kNDVI) (Camps-Valls et al., 2021), and three versions of the land surface water index (LSWI) (Gao, 1996; Xiao et al., 2004), each based on a different shortwave infrared band (centered at 1240, 1640, and 2130 nm, respectively). EVI, which includes soil and atmospheric adjustment factors, is generally more robust

to soil background reflectance than NDVI (Huete et al., 1994, 2002). The relatively new $NIR_v$ and kNDVI have not yet been widely tested, but $NIR_v$ has shown strong performance for predicting seasonal variability of dryland GPP (Wang et al., 2022).

LSWI is related to water content of the land surface due to the strong absorption of shortwave-infrared by water (Ceccato et al., 2001; Gao, 1996; Xiao et al., 2004).

We also obtained 1-km resolution LST estimates for each site from thermal infrared observations onboard both MODIS Terra (MOD11A1) and Aqua (MYD11A1) (Wan, 2014), derived via a view-angle-dependent split window algorithm (Wan and

Dozier, 1996). Each product provides one nighttime and one daytime observation per day (Table 1), with local overpass times at approximately 10:30 am and 10:30 pm (Terra) and 2:30 am and 2:30 pm (Aqua). For regional prediction, we used the corresponding 0.05° resolution CMG product (MOD11C1/MYD11C1). Importantly for dryland ecosystems, accuracy of the MOD11/MYD11 collection 6 LST retrievals over bare soil is considerably improved compared to previous versions due to inclusion of separate daytime and nighttime coefficients and an emissivity adjustment model (Wan, 2014).


To capture soil moisture and soil temperature, we used daily (0:00 UTC retrieval) 9-km resolution surface (0-5 cm depth) and rootzone (0-100 cm) soil moisture and soil temperature estimates from the SMAP Level 4 Soil Moisture (L4SM) "analysis update" product (Reichle et al., 2019) (Table 1). SMAP L4SM assimilates satellite-observed L-band (1.41 GHz) microwave brightness temperature (sensitive to moisture in the upper layers of the soil and vegetation) into a hydrological model forced

with instrumental precipitation observations (Reichle et al., 2019). While soil moisture (rather than soil temperature) is more directly related to the L-band microwave signal and is the primary retrieval objective for L4SM (Reichle et al., 2017), we also chose to use soil temperature estimates due to the strong dependence of soil respiration (and thus NEE) on soil temperature (Curiel Yuste et al., 2007) and its use in other SMAP-based carbon models (Jones et al., 2017). For site-level calibration and evaluation, we retrieved soil moisture and temperature from the grid cell nearest to each tower. We note, however, that unlike

MODIS resolutions (500 or 1,000 meter), the 9-km SMAP resolution is much coarser than the typical ~1 $km^2$ (or smaller) eddy covariance footprint (Chu et al., 2021), so the SMAP soil moisture/temperature estimates used here represent a larger area-integrated average that may not be perfectly representative of conditions inside the flux footprint. For regional analyses, we resampled the SMAP L4SM products to match the 0.05° MODIS CMG resolution using the nearest neighbor. In 2019, SMAP went into "Safe Mode" from June 19 through July 23 (Reichle et al., 2022), during which the L4SM model could not assimilate

microwave brightness temperature and model estimates would have come solely from the hydrological model forced with meteorological observations. Because this only affects two partial months, we chose to retain soil moisture and temperature estimates during this period, though this may result in slightly higher error or bias in our monthly DrylANNd carbon and water flux estimates for June and July 2019.

Since the relationships between vegetation indices and ecosystem function can vary among different vegetation types (Wang et al., 2022), we used the 2020 fractional cover of annual and perennial grasses and forbs, trees, shrubs, litter, and bare ground

from Rangeland Analysis version 3 (Jones et al., 2018; Allred et al., 2021) as static predictors. Rangeland Analysis cover fuses Landsat and MODIS surface reflectance at ~30-meter resolution. For site-level model development, we averaged the fractional covers for all pixels within a 500-meter buffer around each eddy covariance tower. For regional analysis, we reprojected and resampled the 30-meter fractional cover to 0.0005° (~50 meter) resolution and then averaged the fractional cover of the 10,000 pixels falling within each 0.05° pixel of the MODIS CMG grid.

From the daily MODIS and SMAP observations, we developed monthly composites of each variable. For the optical vegetation "greenness" indices (NDVI, kNDVI, EVI, and NIR$_v$), we used maximum value compositing (Holben, 1986) of the valid daily observations within each month, consistent with theoretical and observational evidence that sources of noise in remotely sensed imagery (e.g., clouds, snow, and atmospheric effects) tend to reduce, not increase, the apparent greenness of the land surface (Viovy et al., 1992; Huete et al., 2002). For LST, LSWI, and SMAP, we averaged the daily observations within each month. In all cases, we only used monthly composites where at least 25% of daily observations were valid within the composite window (i.e., at least eight days of valid observations within a given month).

## 2.3 Model framework and initialization

We used feed-forward artificial neural networks (ANNs; Fig. 2) to jointly predict monthly GPP, NEE, and ET. ANNs are effective at finding underlying relationships within multidimensional and multisource datasets, including nonlinear relationships and interactions among predictor variables (Olden et al., 2008). They are particularly useful for estimating biophysical parameters because they support non-linearity, adaptivity to changes in the environment, and decision confidence (Mas and Flores, 2008; Jensen et al., 2009). Synthetic "neurons," where each neuron is a mathematical function, connect the neural network's input and output layers, often through "hidden" layers of intermediary functions. Importantly, ANNs are appropriate for multi-output regression problems, where a single model simultaneously produces predictions of multiple variables (e.g., Atkinson and Tatnall, 1997). Because the multi-neuron output layer of the neural network allows joint prediction of response variables, the ANN framework therefore implicitly preserves some biophysical connections between GPP and NEE, where GPP is the carbon input into the ecosystem, and between GPP and ET, which are coupled via plant stomata.

The DrylANNd model consists of an ensemble of ANNs, each with one input layer of 20 "neurons" (i.e., seven optical VIs, LST observations from four different times per day, three SMAP variables, and six static fractional cover classes), two hidden layers, and one output layer with three neurons (GPP, NEE, and ET). The sizes of the two hidden layers (L$_1$ and L$_2$) were determined based on the number of neurons in the input ($N$) and output ($m$) layers (Huang, 2003):

$$L_1 = \sqrt{(m + 2)N} + 2\sqrt{N/(m + 2)} = 14$$
$$L_2 = m\sqrt{N/(m + 2)} = 6$$

Each ANN in the ensemble (§2.4 below) was initiated with randomly assigned weights and biases based on the Nguyen-Widrow method (Nguyen and Widrow, 1990) and with different random subsets of observations for model training (75%) and validation (25%), with the precise number of data points used for each individual ANN varying slightly depending on the length of the withheld site's data record. We trained the ANNs using the Levenberg-Marquardt algorithm, which performed faster than and at least as well as other training algorithms in early tests.

## 2.4 Model calibration, evaluation, and prediction

Our DrylANNd model consists of an ensemble of 560 individual ANNs, where each ANN in the ensemble was trained with a different combination of sites (always withholding data from one site to use for independent evaluation) and initialized with different weights and biases that connect one layer to the next. Specifically, we withheld each AmeriFlux site from model development for 20 of the ensemble members (28 sites × 20 models per site = 560 total models), training the ANNs for those ensemble members using the remaining 27 sites. We then predicted GPP, NEE, and ET for the withheld site using each of the 20 ensemble members, with the ensemble mean used as the best estimate of GPP, NEE, and ET. This ensures that model skill is assessed based on observations from a site that was new to the model, thus providing a representative measure of the model's ability to extrapolate to locations on which it was not trained. Using the full ensemble of 560 ANNs, we predicted monthly GPP, NEE, and ET using the 0.05° resolution remotely sensed data for all Western U.S. drylands from April 2015 (the first full month of SMAP data) through December 2020. We used the ensemble mean at each pixel as an estimate of GPP, NEE, and ET, with the $10^{th}$-$90^{th}$ percentiles used as estimates of uncertainty.

We evaluated model skill based on the coefficient of determination ($R^2$) and mean absolute error (MAE) between model predictions and observations at each site, using only model predictions generated from the ensemble members in which that site was withheld from training. In addition to evaluating skill at monthly resolution, we also assessed DrylANNd's ability to capture both the mean seasonality (i.e., the mean monthly fluxes during the 2015-2020 period) and the (inter)annual variability at the 16 AmeriFlux sites that were operational over the full SMAP period (2015-2020). We assessed the ability of the model to capture the mean seasonal cycle using both the $R^2$ and the standard deviation ratio (SDR), i.e., the ratio of the standard deviations of the modeled mean seasonal cycle and the observed mean seasonal cycle (Smerdon et al., 2011), where SDR<1 indicates that the model underestimated the seasonality and SDR>1 indicates that the model overestimated the seasonality. To assess the ability of the model to capture (inter)annual variability, we calculated total annual fluxes (i.e., the mean daily flux multiplied by the number of days) in each of the six study-period years during the April-October warm season, with interannual variability defined by comparing the ability of the DrylANNd model to capture variance in flux *anomalies* (i.e., the departure of each year's flux from that site's study-period mean flux) across all sites. We specifically examined annual warm season, rather than calendar year, fluxes because 1) SMAP began operation on March 31, 2015 so there is not a continuous record of January-March fluxes over the full 2015-2020 period and 2) many high elevation and/or high latitude sites have extensive missing data during the cold season due to snow cover. As a benchmark for model skill, we also compared DrylANNd

predictions to the MODIS GPP (MOD17A2HGF; Running et al., 2004) and ET (MOD16A2GF; Mu et al., 2007, 2011) products.

The "black box" nature of many machine learning methods (including ANNs) typically makes it challenging to examine the effect of any given input variable on model predictions. Here, we examined the importance of the MODIS and SMAP predictor variables for model skill in two ways. First, using the same leave-one-site-out calibration and evaluation procedure described above, we ran models based on each of the three classes of variables (MODIS VIs, MODIS LST, and SMAP soil moisture/temperature) individually and in all possible combinations and compared the predictive ability ($R^2$) of each model to

the combined model with all three classes of variables together. Second, we tested the leverage of each time-varying predictor variable by repeatedly (100 times) randomly permuting each variable (thus destroying its information content) and re-running model predictions, similar to established perturbation and stepwise methods for uncovering the most critical variables in ANNs (Gevrey et al., 2003). The synthetic, noise-only permutations of each variable were drawn from a normal distribution with the same mean and variance as observed for that variable at that site. We calculated both the mean percent increase in MAE and

the change in $R^2$ when each variable was replaced with noise-only permutations, thus estimating its leverage on model skill.

## 3 Results

### 3.1 Overall performance

At the monthly scale, the DrylANNd model explained more than 70% of the combined spatial and temporal variation in GPP (Fig. 3a) and ET (Fig. 3e) for sites withheld from model training, but only about 35% of the variation in NEE (Fig. 3c). For

GPP, the model performed best at shrubland sites ($R^2$~0.8), followed by savanna and grassland sites ($R^2$~0.7), and evergreen needleleaf sites ($R^2$~0.6) (Fig. 3b). However, while the model performed worst on average at evergreen needleleaf sites, where optical VIs struggle to capture seasonal dynamics in GPP (Wang et al., 2022), these sites also saw the greatest *improvement* over MODIS GPP estimates ($R^2$~0.6 vs. $R^2$~0.25) (Fig. 3b). DrylANNd outperformed MODIS GPP estimates at 19 sites, while MODIS outperformed DrylANNd at 7 sites (with nearly identical performance at two sites). Overall, however, DrylANNd

showed improvement over MODIS for all four vegetation types (Fig. 3b). Model skill was considerably worse for NEE (Fig. 3c-d), with best performance at shrubland sites ($R^2$~0.7), followed by grasslands ($R^2$~0.4), and then by savannas ($R^2$~0.3) and evergreen needleleaf forests ($R^2$~0.1) (Fig. 3d). The DrylANNd model particularly excelled at capturing variation in ET (Fig. 3e-f), especially in shrublands ($R^2$>0.8), with $R^2$>0.6 on average across all vegetation types (Fig. 3f). Site-level $R^2$ varied from ~0.4-0.9, which represents an improvement (in many cases substantial) over MODIS ET estimates at 23 out of 26 sites and for

all four vegetation types (Fig. 3f).

DrylANNd also effectively captured spatial variation of warm-season carbon and water fluxes across western U.S. drylands (Fig. 4). The model simulates realistic spatial gradients of GPP (Fig. 4a), NEE (Fig. 4c), and ET (Fig. 4e), with highest

productivity and ET in the subhumid east and in high elevation "sky islands," where cooler temperatures and more abundant precipitation provide a more favorable environment than the surrounding desert lowlands. Across the 16 AmeriFlux sites that completely overlap the SMAP observational period, DrylANNd captured 75-80% of the spatial variation in warm-season GPP and ET (Figs. 4b and 4f, respectively) with minimal bias (i.e., with predictions all falling along the 1:1 line). For NEE, the model captured ~50% of the spatial variation but with a negative bias indicating an overestimation of the carbon sink across the region, particularly for evergreen needleleaf forest sites (Fig. 4d).

## 3.2 Seasonality

At the 16 AmeriFlux sites that cover the full SMAP period (2015-2020), DrylANNd effectively captured the mean seasonality (i.e., mean monthly fluxes) of both GPP and ET across most sites, with $R^2 \geq 0.8$ at all but two sites for both GPP (Fig. 5) and ET (Fig. 6). This represents an improvement over MODIS GPP seasonality at 12/16 sites and MODIS ET at 13/16 sites. Likewise, DrylANNd effectively represented the seasonality of GPP and ET, with SDR closer to 1 at 10/16 sites for GPP (Fig. 5) and 15/16 sites for ET (Fig. 6). However, even though DrylANNd's ET estimates come considerably closer to representing the seasonality of ET, they still underestimate the magnitude of ET seasonality (SDR<1) at all but two sites (US-Mpj and US-Ses; Fig. 6), suggesting that the disproportionately-important "hot moments" of dryland water fluxes that occur during intermittent pulses of rainfall are still not completely captured by the model. Likewise, DrylANNd often misses the bimodal spring/summer growing seasons at many of the North American monsoon-influenced sites (Figs. 5-6, S1-S2). For example, DrylANNd correctly estimated the magnitude of monsoon-driven summer GPP and ET but underestimated the spring GPP and ET at the Santa Rita Grassland (US-SRG) site in southeastern Arizona (Fig. 5-6, S1-S2), and it missed the mid-summer suppression of GPP at the two Valles Caldera sites (US-Vcp and US-Vcm) in New Mexico (Fig. 5, S1).

While the seasonality of NEE was mostly well captured by DrylANNd ($R^2 \geq 0.6$ at most sites; Fig. S3), it tended to systematically underestimate NEE (i.e., overestimate the magnitude of the carbon sink) at several sites (Fig. S4), particularly the three forest sites (US-Me6, US-Vcm, and US-Vcp), with a smaller number of sites where NEE was slightly overestimated. In the most extreme case (US-Vcp), the mean bias in NEE exceeded $-1$ g C m$^{-2}$ day$^{-1}$, with biases approaching or exceeding $-2$ g C m$^{-2}$ day$^{-1}$ during the June-September period when measured NEE at the site was near zero.

## 3.3 (Inter)annual variability

DrylANNd captured roughly 70% of the variability in annual warm-season GPP (Fig. 7a) and 66% of the variability in warm-season ET (Fig. 7c) with MAE of ~100 g C m$^{-2}$ for GPP and ~50 mm for ET, a considerable improvement over the MODIS GPP and ET estimates. However, much of this skill is likely attributable to strong performance at estimating spatial (rather than temporal) variation across the study sites (Fig. 4 and §3.1); like many remote sensing estimates of GPP and ET (Smith et al., 2019; Biederman et al., 2017; Stocker et al., 2019), DrylANNd struggled to capture the interannual *variability* (i.e., deviations from site mean) of carbon and water fluxes. DrylANNd only captured 31% of the interannual variability of GPP

(Fig. 7b), similar to that of MODIS GPP ($R^2$=0.33) though with a slope slightly (but not significantly) closer to 1. Like MODIS, DrylANNd also underestimated the magnitude of interannual variability of ET, with a slightly lower $R^2$ than MODIS but a slope slightly closer to 1 (Fig. 7d).

### 3.4 Variable importance

For all three response variables (GPP, NEE, and ET), models that included all three subsets of predictor variables (optical VIs, LST, and SMAP) performed best overall (Fig. 8a-c), though a combination of optical vegetation indices with SMAP soil moisture/temperature performed nearly as well for both GPP (Fig. 8a) and ET (Fig. 8c). Models based on VIs and/or LST performed worse than SMAP-based models but still achieved overall $R^2$ of ~0.4 for GPP and ~0.5 for ET (compared to $R^2$~0.7 for the models that included all predictor variables). The same was generally true for the models' ability to capture within-site

temporal variability (i.e., anomalies relative to monthly site-means; Fig. 8d-f). For both GPP (Fig. 8d) and ET (Fig. 8f), the model based on all predictor variables performed best for capturing temporal variability, followed closely by the VI+SMAP model.

Compared to models based solely on optical VIs, the addition of SMAP soil moisture/temperature generally made the largest

difference for model performance in grasslands and shrublands, while including LST estimates from MODIS thermal infrared made the largest difference for model performance in evergreen needleleaf forests (Fig. 8). In shrublands, the $R^2$ of VI+SMAP GPP (Fig. 8a) and ET (Fig. 8c) increased by more than 0.1 compared to the VI-only models, and it increased by more than 0.2 for NEE (Fig. 8b). In grasslands, the difference between the VI-only and VI+SMAP models was even larger, with $R^2$ increasing by more than 0.2 for all three flux variables (Fig. 8a-c). Generally, the addition of thermal data, however, offered little (if any)

gains in model performance over either the VI-only or VI+SMAP models for either grassland or shrubland sites (Fig. 8a-c), though it did improve representation of flux *anomalies* at grassland sites compared to VI-only models (Fig. 8d-f). Conversely, across evergreen needleleaf forest sites, the VI+SMAP models performed slightly worse than the VI-only models for predicting monthly fluxes (Fig. 8a-c), suggesting that models trained on other evergreen needleleaf sites were overfit to the SMAP data and were less skillful at extrapolating to an "unseen" site than a model based solely on optical VIs. However, including SMAP

soil moisture/temperature did slightly improve the ability of the models to predict monthly flux *anomalies* at evergreen needleleaf forests (Fig. 8d-f). Including thermal data, on the other hand, improved predictions of both monthly fluxes and monthly flux anomalies in evergreen needleleaf forests compared to VI-only models. For savanna sites, both LST and SMAP soil moisture/temperature improved predictions of monthly fluxes compared to VI-only models (Fig. 8a-c), though SMAP was far more important for predicting flux anomalies (Fig. 8d-f).

While the LST-only models performed the worst overall (Fig. 8), the models based on all variables assigned high leverage to LST for all three response variables, wherein a random permutation of LST led to large increases in MAE and decreases in $R^2$ (Fig. 9). For GPP, the four diurnal LST observations (particularly daytime Terra LST), along with NDVI and $NIR_v$, tended to

have the highest "leverage" on model skill (Fig. 9a-b). There was little variation in leverage among the predictor variables for
NEE (Fig. 9c-d), with random permutations generally leading to relatively low and site-specific changes in both MAE and $R^2$.
Nighttime LST observations had the greatest leverage over the MAE of NEE predictions (Fig. 9c), while NDVI/NIR$_v$ and all
four daily LST observations had the most leverage over predictive $R^2$ (Fig. 9d). For ET, a combination of all three classes of
variables (vegetation indices, LST, and soil moisture/temperature) contributed positively to model skill, with NDVI/NIR$_v$,
daytime and nighttime Terra LST, and SMAP surface soil moisture generally holding the highest leverage over model skill
(Fig. 9e-f).

## 4 Discussion

### 4.1 Model skill, strengths, and shortcomings

Here, we developed and evaluated a data-driven, machine learning-based approach for estimating monthly carbon (GPP, NEE)
and water (ET) fluxes in U.S. drylands using multi-source satellite remote sensing. Our DrylANNd model incorporated
information from the optical, thermal, and microwave domains, including newer optical VIs that have shown promise in
drylands (i.e. NIRv; Wang et al., 2022), daily land surface temperature observations from multiple times per day, and estimates
of surface/rootzone soil moisture and soil temperature. DrylANNd performed particularly well at monthly and seasonal (i.e.,
mean monthly) time scales, representing a considerable improvement over MODIS GPP and ET estimates across most eddy
covariance sites and all vegetation types (Fig. 3).

DrylANNd particularly excelled at capturing monthly (Fig. 3e-f), seasonal (Fig. 6), and spatial (Fig. 4e-f) variation in ET.
Given the importance of ET for linking the carbon, water, and energy cycles (Fisher et al., 2017), accurate ET estimates are
critical for understanding and monitoring global ecosystem function, especially in drylands where remote sensing of ET is
particularly challenging (Smith et al., 2019; Fisher et al., 2017). By contrast, NEE proved more challenging to estimate than
either GPP or ET (Fig. 3c-d, 4c-d), likely because many processes involved in ecosystem respiration cannot be easily
represented with satellite data. While heterotrophic and autotrophic respiration rates are strongly dependent on temperature
(Atkin and Tjoelker, 2003) and soil moisture (Moyano et al., 2013), which can be captured by MODIS LST and SMAP soil
moisture/temperature, they also depend on microbial community composition, substrate availability, and root biomass that are
not visible to satellite sensors. DrylANNd performed moderately well at capturing the seasonality (Fig. S3) and spatial
variation (Fig. 4c-d) in NEE but tended to systematically overestimate the magnitude of net carbon uptake in U.S. drylands
(Fig. 4d, S3), particularly in evergreen needleleaf forests. Many dryland sites have a net carbon balance near zero and can flip
between being sources of and sinks for $CO_2$ in any given year (Biederman et al., 2018; Scott et al., 2009, 2010, 2015), so even
small errors or biases in NEE estimates can have large effects on dryland carbon budgeting and monitoring.

While DrylANNd captured monthly, seasonal (i.e., mean monthly), and spatial variation of GPP and ET with fidelity, it struggled to predict interannual variability (Fig 7). This is a common issue for satellite-based models applied in dryland ecosystems (Biederman et al., 2017; Smith et al., 2019; Stocker et al., 2019; Barnes et al., 2021), partly due to the prevalence of "hot moments" (i.e., short periods of high biogeochemical activity) that are disproportionately important to time-averaged carbon and water fluxes in drylands (Kannenberg et al., 2020). While DrylANNd has relatively little systematic bias at

capturing low extremes in monthly GPP (Fig. 3a) and ET (Fig. 3e), it tended to underestimate the high extremes. DrylANNd's monthly resolution may smooth the intensity of these short but impactful "hot moments," leading to systematic underestimation of monthly high extremes which also propagates to longer timescales, with DrylANNd clearly underestimating the high extremes in interannual variability of warm season GPP (Fig. 7a) and ET (Fig. 7c). Improving estimates of interannual variability of dryland systems may therefore require models that operate at finer temporal resolutions (e.g., daily) to adequately

represent short, intense periods of pulse-driven dryland vegetation activity.

    Despite the challenges in capturing interannual variability, the ANN machine learning approach used here has several key benefits. First, because it is a data-driven model based solely on remote sensing products with short latencies, DrylANNd would be relatively easy to operationalize at a large scale and in near real time. Second, the ensemble approach allows for

intuitive estimates of uncertainty, which are critical for many applications (e.g., ecological forecasting) but which are rarely provided (Dietze et al., 2018). Finally, neural networks allow joint modeling of multiple response variables, providing the means both to efficiently generate multiple indicators of ecosystem activity and to partially preserve the physical connections between GPP and NEE and between GPP and ET, which is relatively rare for remote sensing-based models. The MODIS and SMAP carbon products, for example, provide joint estimates of GPP and net primary production (Running et al., 2004) and

GPP and NEE (Jones et al., 2017), respectively, but neither provides estimates of ET that are coupled to GPP. Zhang et al. (2016), on the other hand, provide coupled estimates of GPP and ET using static, biome-specific water-use efficiencies, but this approach does not provide estimates of downstream plant or ecosystem carbon balances, nor does the model allow for the dynamic changes in water-use efficiency that can occur in response to pulses of rainfall or variation in vapor pressure deficit (Roby et al., 2020).

**4.2 Benefits of multi-source remote sensing**

    Previous work has highlighted the potential for combining multiple remote sensing proxies to improve the representation of vegetation dynamics (Stavros et al., 2017; Smith et al., 2019), and our results support this conclusion and provide further guidance on which remotely sensed variables contribute most to model improvement in drylands. All three classes of remote sensing variables (optical, thermal, and microwave) contributed positively to model skill (Figs. 8-9). In particular, the inclusion

of SMAP soil moisture/temperature resulted in large gains in model skill (Fig. 8), with the VI+SMAP models performing substantially better than the VI-only models, especially for GPP (Fig. 8a), for which the monthly $R^2$ improved from ~0.4 to nearly 0.7 when adding SMAP variables as predictors. This is consistent with site-level research showing the importance of

soil moisture for dryland carbon and water fluxes (Novick et al., 2016; Stocker et al., 2018). The inclusion of soil moisture and soil temperature was particularly important for predicting carbon and water fluxes in grassland, shrubland, and savanna systems.

While the LST-only models usually performed worst of all model subsets, and the monthly VI+LST models barely outperformed the VI-only models (Fig. 8a-c), the inclusion of LST was particularly useful for improving predictions of monthly *anomalies* (i.e., deviations from the mean monthly flux of a given site) of all three flux variables, with $R^2$ increasing by ~0.1 when adding LST to the VI-only models (Fig. 8d-f). When LST was included in the models, it also had very high leverage over model skill (Fig. 9), indicating that it provided unique information not captured by other remote sensing sources. Among the different vegetation types, LST was particularly important for evergreen needleleaf forests (Fig. 8), where optical VIs struggle to capture both spatial and temporal dynamics of GPP (Wang et al., 2022) and where growing seasons are more likely to be limited by low temperatures than in warmer and drier grasslands, shrublands, and savannas. Daytime LST was particularly important for predicting GPP and ET, possibly indicating that it is an effective indicator of moisture stress (Javadian et al., 2022; Still et al., 2021), while nighttime LST was particularly important for predicting NEE, consistent with previous work showing the importance of nighttime temperature for respiration and overall carbon balance (Anderegg et al., 2015). LST also has the benefit of a much longer operational record and much finer spatial resolution than SMAP, so models based on a combination of VIs and LST could allow longer records of dryland responses to hydroclimatic variability and change.

Somewhat surprisingly, NDVI held higher leverage over model predictions than most other predictor variables (Fig. 9), despite previous research documenting significant flaws in its ability to track dryland GPP (Wang et al., 2022; Yan et al., 2019). The relatively new NIR$_v$ index also held very high leverage, consistent with recent work showing that it is particularly effective at tracking GPP variation in low productivity, sparsely-vegetated grasslands and shrublands (Wang et al., 2022). EVI, on the other hand, was assigned very low leverage across nearly all sites and all fluxes, despite generally being regarded as an improvement over NDVI in drylands due to the inclusion of a soil background adjustment factor (Smith et al., 2019). This could plausibly result from EVI containing similar information as other optical vegetation indices and thus being assigned less weight in the models. Wang et al. (2022), for example, found that the soil-adjusted vegetation index (SAVI), on which EVI was partly based (Huete et al., 1994, 1997, 2002), performed very similarly to but slightly worse than NIR$_v$ for tracking seasonality and spatial variability of dryland GPP.

### 4.3 DrylANNd applications and priorities for future dryland model development

Given the challenges of mitigating and adapting to a changing climate, high-quality remotely sensed carbon and water flux estimates are needed for large-scale monitoring of changes in global ecosystem function and ecosystem services, especially in dryland regions that are warming more rapidly than many other regions (Huang et al., 2017). Ecosystem production estimates

provide the means to monitor and forecast rangeland and cropland productivity (e.g., Hartman et al., 2020) and to track changes in the terrestrial carbon cycle (Xiao et al., 2019). Evapotranspiration estimates are needed for monitoring drought and plant water use and water stress (Fisher et al., 2017), which in turn affect both fire risk (Rao et al., 2022) and mortality risk (McDowell et al., 2022).

Our DrylANNd approach has significant potential to provide these capabilities in the western U.S. Despite its short calibration and validation period, DrylANNd's training data encompasses much of the climate variability experienced by the western U.S., including both anomalously wet and dry years that may serve as analogues when running the model forward in time as new MODIS and SMAP data are released. However, it is possible that the historically atypical "megadrought" conditions (Williams

et al., 2020, 2022; Dannenberg et al., 2022) under which the model was trained may impose limitations on the model's predictive capability. Some of the model's limitations in capturing interannual variability could perhaps be ameliorated by incorporating additional remote sensing data that capture other aspects of dryland ecosystem function. Solar-induced fluorescence, for example, effectively tracks vegetation activity in drylands (Smith et al., 2018; Wang et al., 2022), particularly in dry evergreen needleleaf forests where reflectance-based optical VIs tend to perform poorly (Magney et al., 2019; Wang et

al., 2022). However, satellite-based SIF estimates suffer from coarse spatial and temporal resolutions (e.g., GOME-2; Joiner et al., 2013), discontinuous spatial coverage (e.g., OCO-2 and OCO-3; Sun et al., 2018), or an even shorter period of record than SMAP (e.g., TROPOMI; Köhler et al., 2018). Fusions of satellite SIF data with MODIS surface reflectance (e.g., Zhang et al., 2018) overcome some of these limitations but would likely inherit many of the same flaws as reflectance-based optical VIs since they are based on the same surface reflectance data. As SIF temporal and spatial resolution improves, it will likely

become increasingly useful for dryland carbon and water modeling. Gravimetric estimates of total terrestrial water storage (Andersen et al., 2005; Humphrey et al., 2018) could also improve the representation of deeper moisture, which can be an essential water source for deep-rooted trees in semiarid systems (Rempe and Dietrich, 2018; McCormick et al., 2021). However, like the longer-term SIF measurements, estimates of total water storage are limited to very coarse (0.5°) spatial resolution and monthly frequency, which would preclude use in higher frequency estimates of carbon and water fluxes.

Applying the DrylANNd approach at a global scale would require expanding the eddy covariance training sites beyond those used here, which are limited solely to western U.S. AmeriFlux sites. Drylands are generally defined as regions where annual precipitation is insufficient to meet evaporative demand (e.g., P/PET < 0.75), but climates falling within that general definition can have very diverse seasonalities, temperatures, and precipitation regimes (e.g., Fig. 1b). A purely empirical, data-driven

model (like DrylANNd) would likely struggle to extrapolate to regions beyond those on which it was trained, especially because dryland carbon and water fluxes tend to be more "unique" to their specific region compared to more mesic systems (Haughton et al., 2018). In addition to improving and expanding the input data, a global-scale version of the DrylANNd approach would therefore benefit from expanding the training network to include dryland eddy covariance sites from other global flux networks (e.g., OzFlux, AsiaFlux, and the various European flux networks).

## 5 Conclusions

Here, we developed and evaluated a machine learning approach (DrylANNd) for jointly modeling of key carbon and water fluxes (GPP, NEE, and ET) specifically for drylands of the western U.S. using a combination of satellite optical vegetation indices, multitemporal thermal infrared, and microwave-based soil moisture and soil temperature. Longstanding challenges in current multispectral satellite-based estimation of dryland carbon and water fluxes are the result of several interacting issues, including poor representation of soil moisture stress, decoupling between "greenness" and plant physiology, high soil background reflectance in open canopies, and limited representation of dryland calibration and validation sites available for model training and testing. Our approach partially addresses these limitations of previous satellite carbon and water flux estimates in drylands:

- Soil moisture is explicitly included in the model rather than relying on the covariance between VPD and soil moisture or water-sensitive vegetation indices as proxies of moisture stress.
- The model includes new vegetation indices (e.g., $NIR_v$) that show promise in capturing dryland seasonality (Wang et al., 2022), along with satellite thermal and microwave observations representing temperature and moisture stress impacts on ecosystem fluxes.
- The model is trained specifically for dryland ecosystems based on an extensive network of 28 eddy covariance sites spanning a large latitudinal and arid-to-subhumid gradient in the western U.S.

We found that this approach effectively captures monthly, seasonal, and spatial variation in GPP and, especially, ET through both space and time, though it still underestimates the magnitude of interannual variability of carbon and water fluxes. DrylANNd was less effective at capturing NEE than GPP or ET, likely because respiratory processes are largely invisible to satellite sensors, with the magnitude of dryland carbon sinks overestimated particularly at evergreen needleleaf sites. Compared to models based solely on optical vegetation indices, the inclusion of SMAP soil moisture/temperature was crucial for improving estimates of both the magnitudes and temporal variabilities of all three fluxes, especially in dry grasslands and shrublands of the western U.S. On the other hand, the addition of multitemporal thermal observations improved flux estimates in evergreen needleleaf forests, where optical vegetation indices have traditionally struggled to capture GPP dynamics. Drylands play important roles both in the global carbon cycle (Poulter et al., 2014; Ahlström et al., 2015) and in ecosystem services supporting a large human population (Reynolds et al., 2007; Bestelmeyer et al., 2015), and DrylANNd significantly improves our ability to quantify carbon and water fluxes in these ecosystems.

**Data and code availability.** Code for all modeling and analysis is available at https://github.com/mpdannenberg/drylANNd, and 0.05° monthly DrylANNd GPP, NEE, and ET estimates for the western U.S. are archived and publicly available as NetCDF files distributed under a creative commons license at Iowa Research Online (https://doi.org/10.25820/data.006185).

**Author contributions.** Conceptualization: MPD, MLB, WKS, RLS, and JAB. Formal analysis: MPD and XW. Funding acquisition: MPD, MLB, WKS, and XW. Methodology: MPD, MLB, MRJ, and SKM. Software: MPD. Visualization: MPD. Writing – original draft preparation: MPD, MLB, and MRJ. Writing – review & editing: all authors.

**Competing interests.** The authors declare that they have no conflict of interest.

**Acknowledgements**. MPD, MLB, WKS, and MRJ were supported by the NASA SMAP Science Team (grant number 80NSSC20K1805), and XW was supported by the NASA FINESST program (grant number 80NSSC19K1335). We thank AmeriFlux and the tower PIs for making eddy covariance data publicly available. Any use of firm, product or trade names is for descriptive purposes only and does not imply endorsement by the U.S. Government. USDA is an equal-opportunity employer and provider.

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

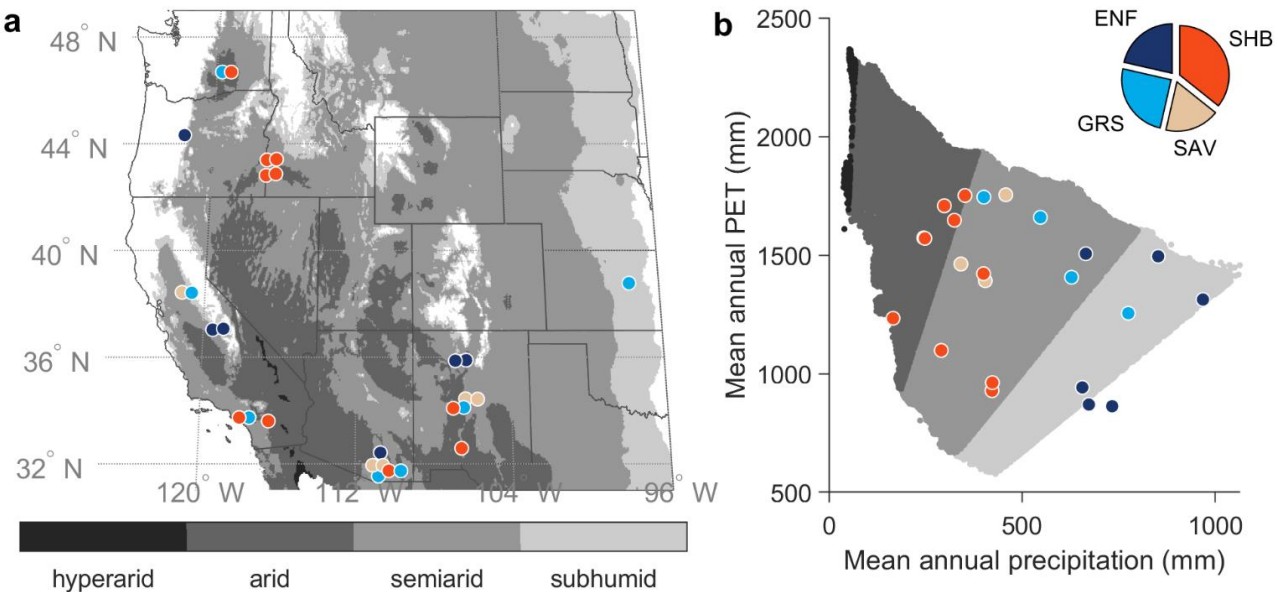

**Figure 1: (a) Map of the geographic domain, aridity, and eddy covariance sites. Adjacent sites were slightly offset to improve visibility. (b) Climatic locations of the eddy covariance sites by mean annual precipitation (P) and potential evapotranspiration (PET). Gray-scale background shows the P and PET distribution of all dryland grid cells shown in (a), while the colored points show the distribution of the AmeriFlux sites in aridity space. (Note that some sites are not visible in (b) because their climates are essentially identical to adjacent sites and their points therefore perfectly overlap.) Aridity classes were defined based on the ratio of**
**mean annual precipitation to potential evapotranspiration (over 1981-2010) from the ~4 km TerraClimate dataset (Abatzoglou et al., 2018): subhumid (0.5 ≤ P/PET ≤ 0.75), semiarid (0.2 ≤ P/PET < 0.5), arid (0.03 ≤ P/PET < 0.2), and hyperarid (P/PET < 0.03). Sites are color coded by vegetation type (shown in the pie chart in (b)): evergreen needleleaf forest (ENF), grassland (GRS), shrubland (SHB), and savanna (SAV). Site details can be found in Table S1.**

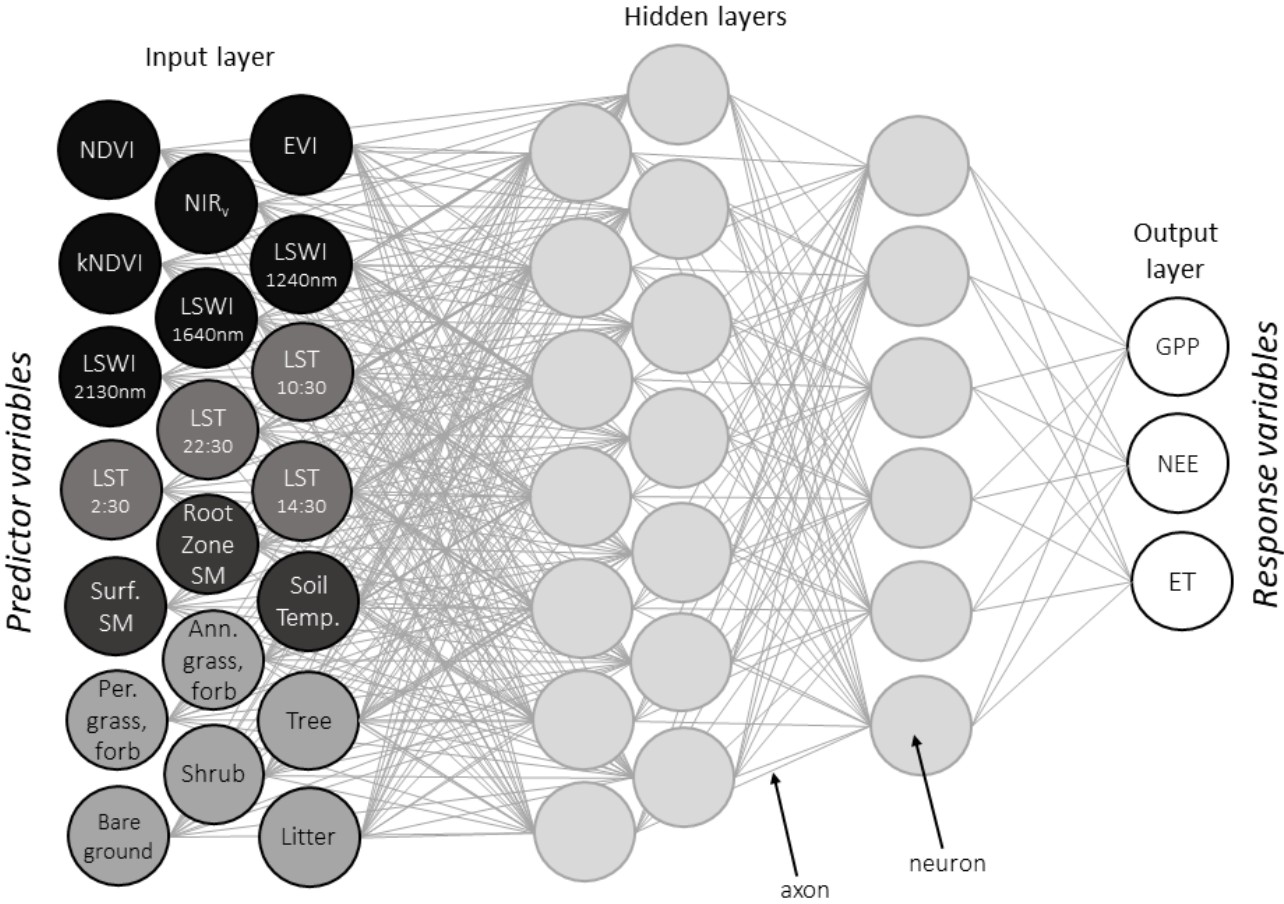

Figure 2: Artificial neural network (ANN) architecture conceptual diagram (Olden et al., 2008). Each "neuron" in a given layer represents a weighted combination of the neurons in the previous layer. All remotely sensed predictor variables in the input layer are also listed in Table 1.

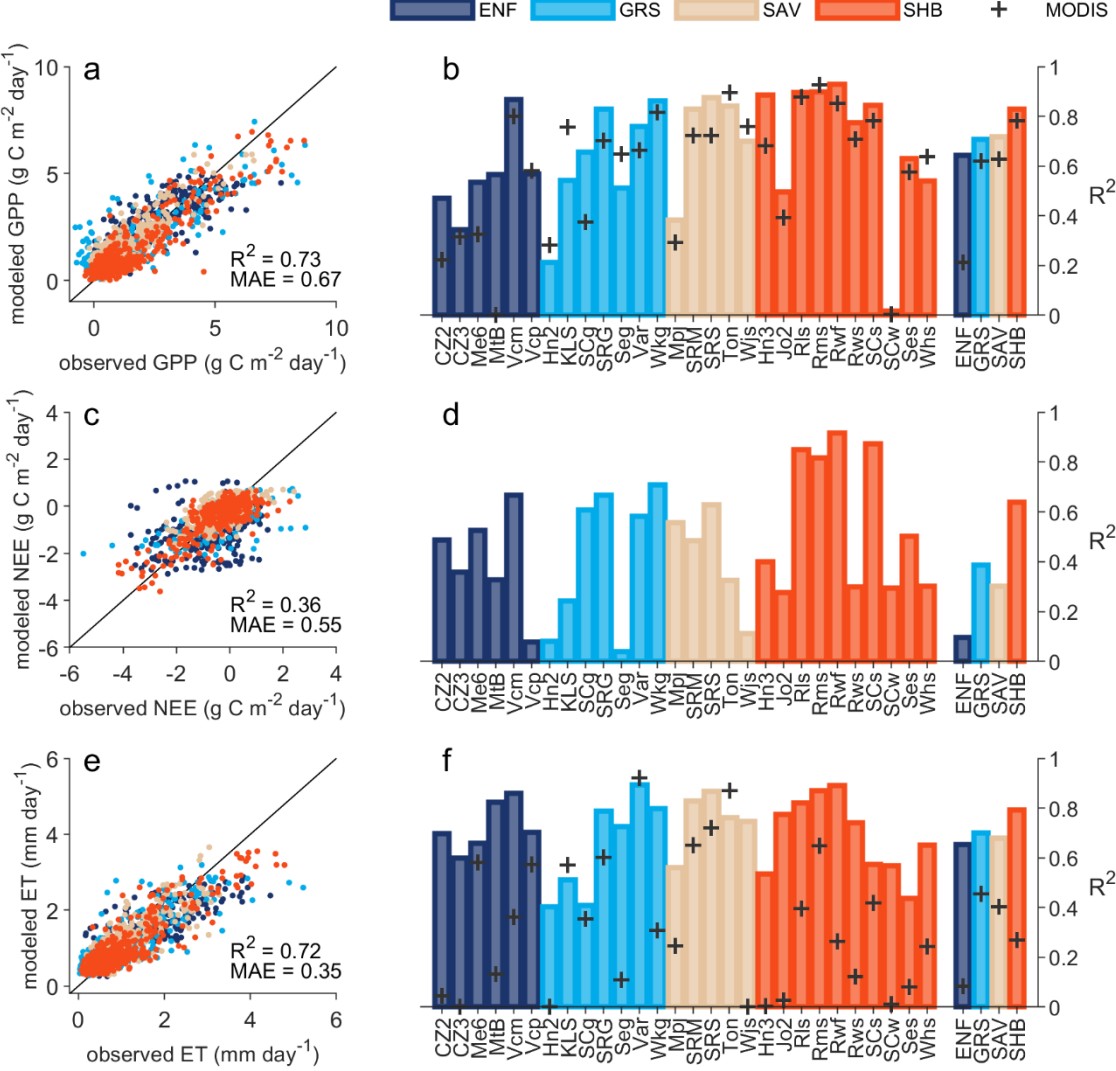

**Figure 3: Overall performance of DrylANNd GPP (a-b), NEE (c-d), and ET (e-f) estimates at monthly time scale. Scatterplots show comparison of monthly tower observations to cross-validated model estimates aggregated across all 28 eddy covariance sites (Table S1). Bar plots show the R$^2$ between monthly tower observations and cross-validated model estimates for each site individually and aggregated by vegetation type (Fig. 1). Gray "+" signs (b & f) show the MODIS product skill for each site and vegetation type.**

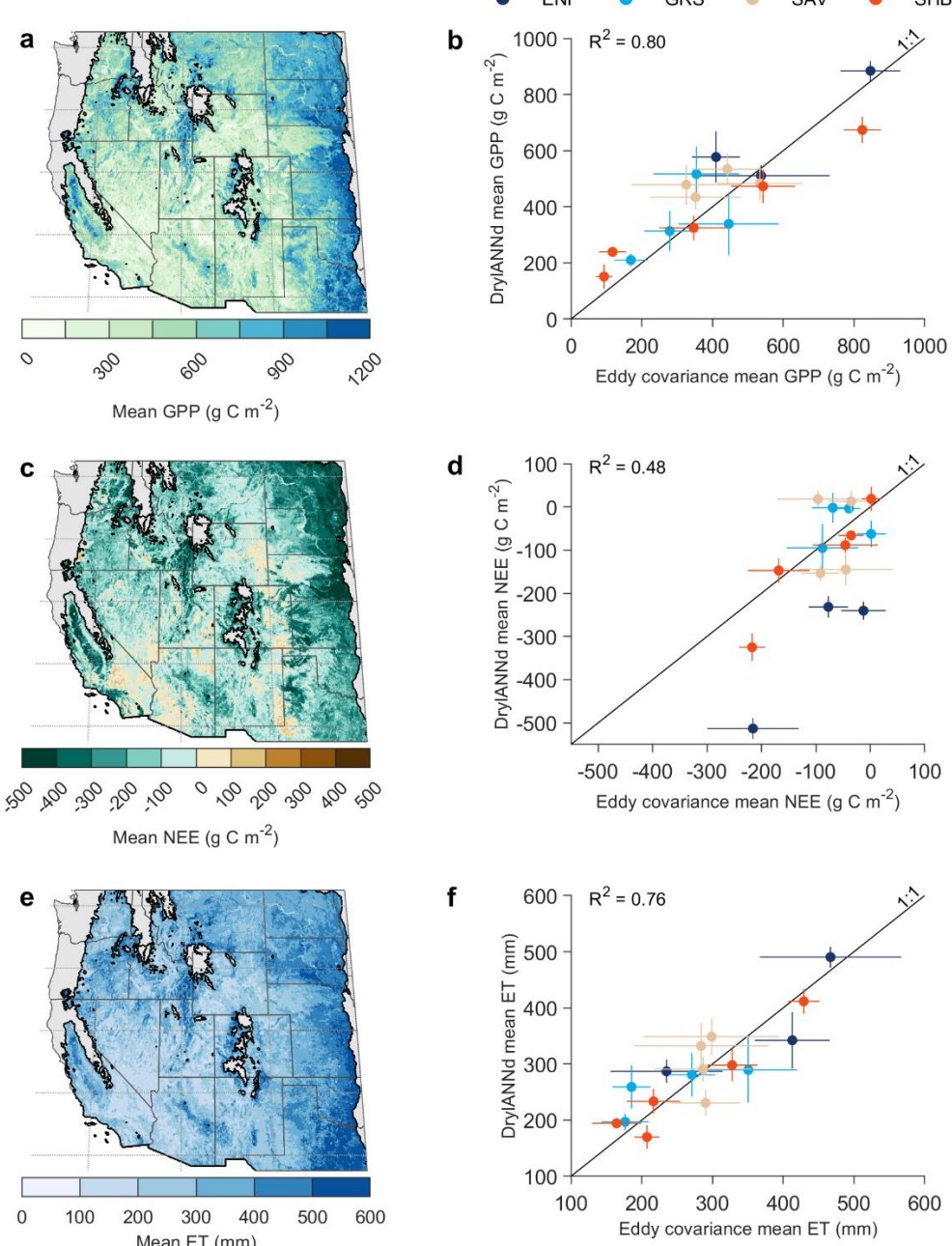


**Figure 4: Spatial patterns and spatial validation of mean warm-season (April-October) GPP (a-b), NEE (c-d), and ET (e-f) predicted by DrylANNd during 2015-2020. Maps show the regional DrylANNd predictions, with the black outline indicating the region classified as dryland (P/PET ≤ 0.75). Scatterplots show mean (±1 standard deviation) warm-season fluxes predicted by DrylANNd and measured by eddy covariance at the 16 AmeriFlux sites that had complete records from 2015-2020, with points colored by vegetation type. For purposes of calculating annual means and standard errors, missing monthly data were filled with a spline interpolation.**


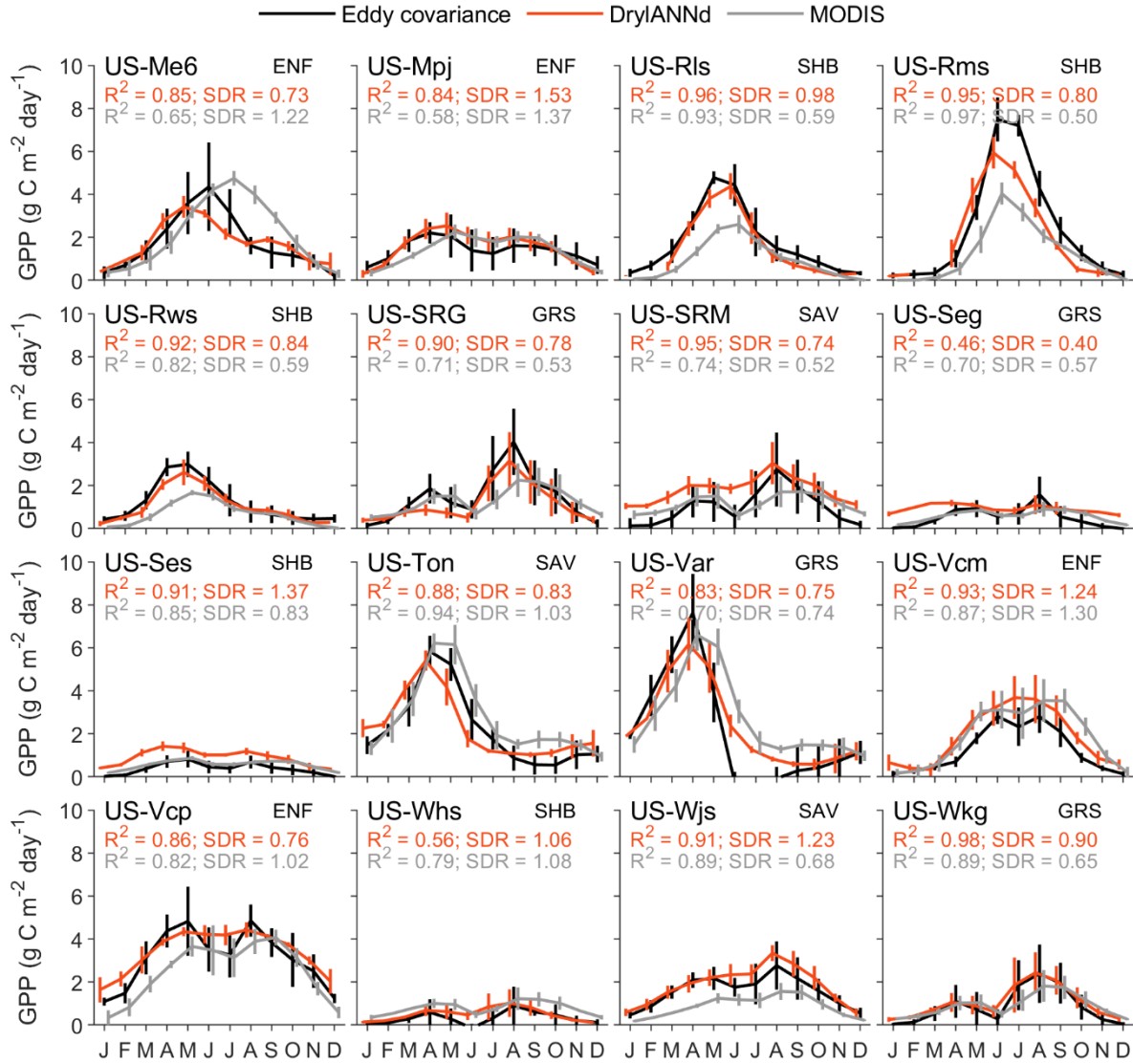

**Figure 5: Mean (±1 standard deviation) seasonality of eddy covariance GPP (black), DrylANNd GPP (red), and MODIS GPP (gray) at the sixteen sites with complete flux records. Monthly means and standard deviations for all three datasets were estimated using all available observations during the period 2015-2020. Standard deviation ratio (SDR)<1 indicates that the model underestimated the magnitude of seasonal variability and SDR>1 indicates that the model overestimated the magnitude of seasonal variability.**

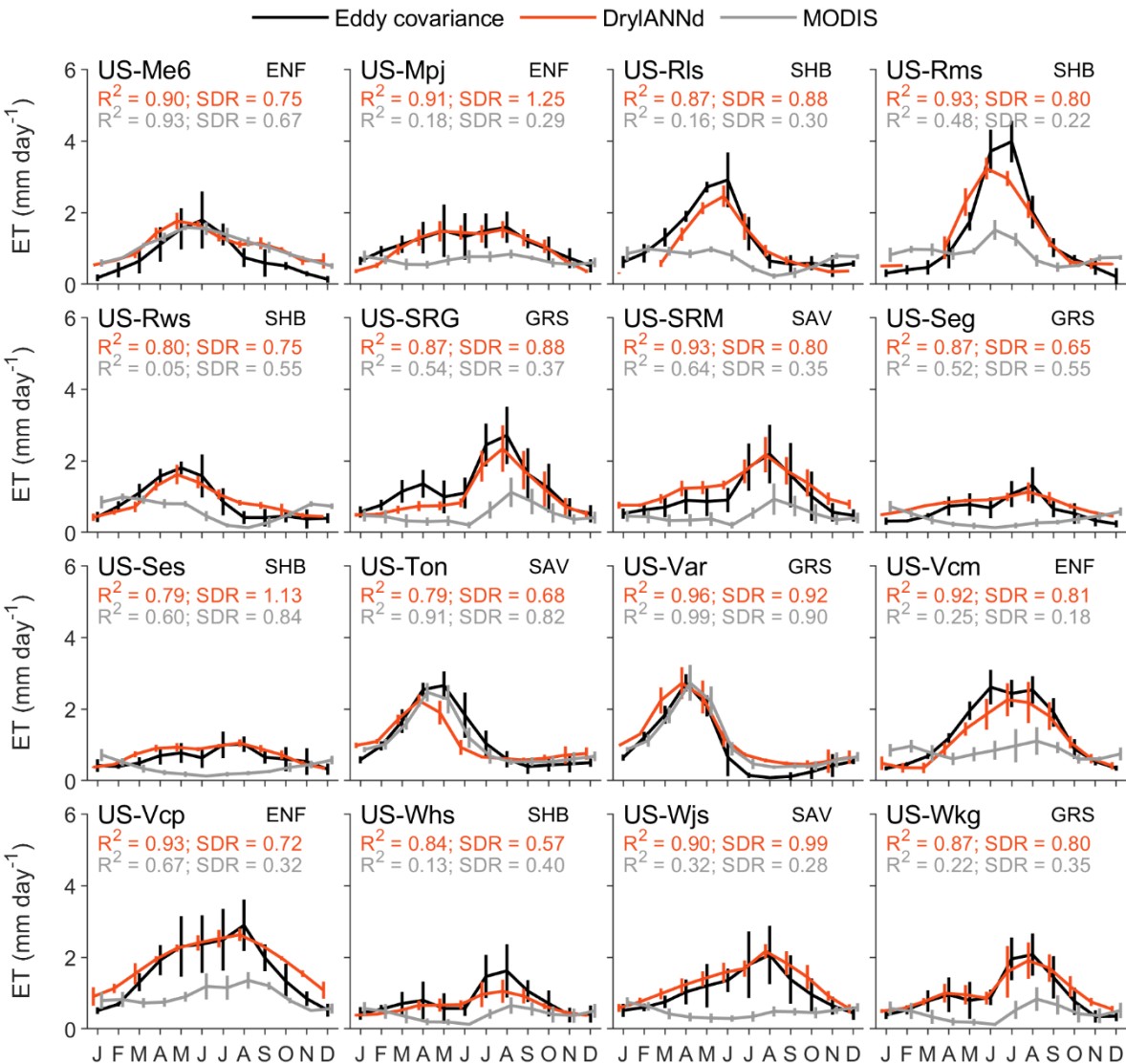

**Figure 6: Mean (±1 standard deviation) seasonality of eddy covariance ET (black), DrylANNd ET (red), and MODIS ET (gray) at the sixteen sites with complete flux records. Monthly means and standard deviations for all three datasets were estimated using all available observations during the period 2015-2020. Standard deviation ratio (SDR)<1 indicates that the model underestimated the magnitude of seasonal variability and SDR>1 indicates that the model overestimated the magnitude of seasonal variability.**

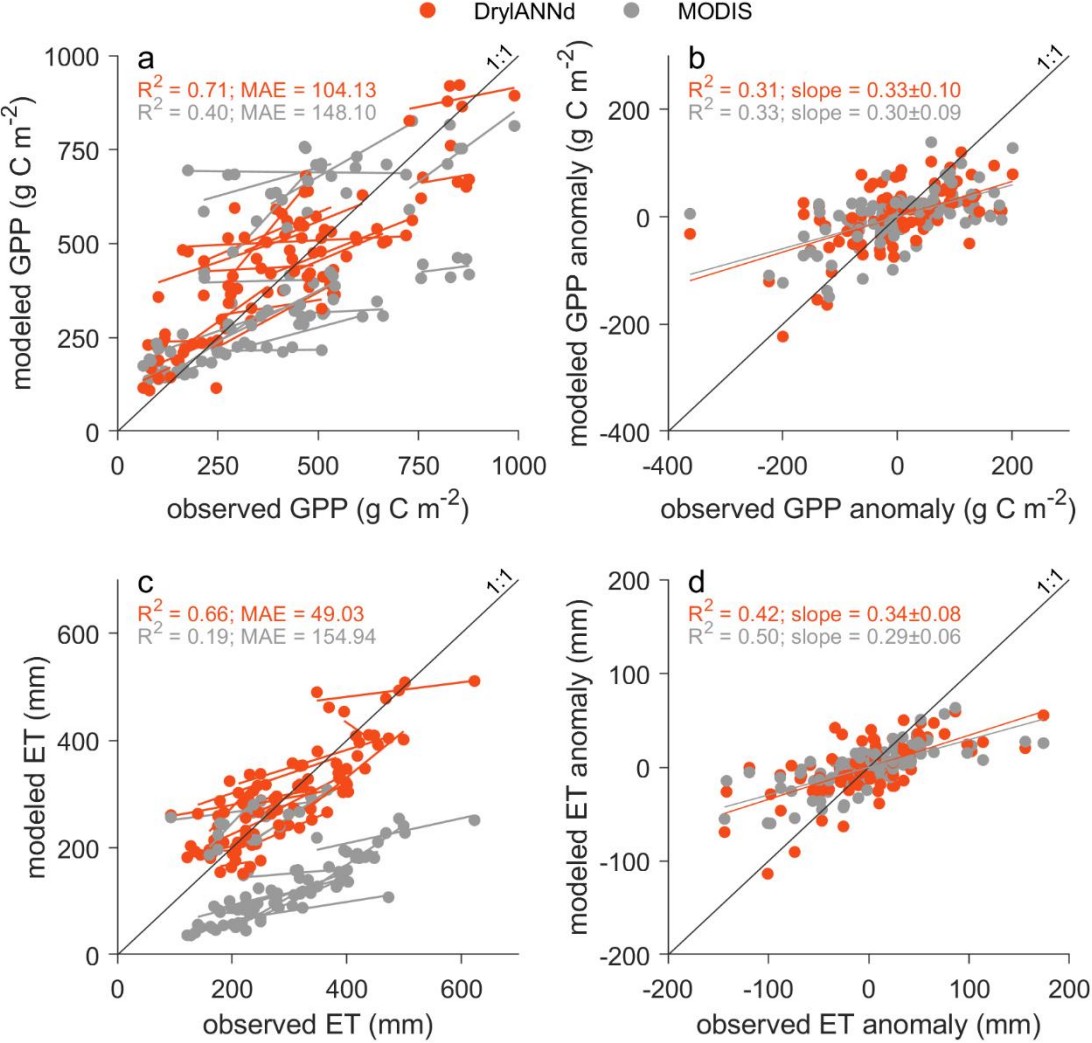

**Figure 7: Annual and interannual variability of warm season (April-October) (a-b) GPP and (c-d) ET. The orange and gray lines in (a) and (c) show the linear relationship between estimated and observed GPP and ET for each individual site during the six-year training and evaluation period, while lines in (b) and (d) show the linear relationship between estimate and observed GPP and ET anomalies across all sites. For purposes of calculating annual values and anomalies, missing monthly data were filled with a spline interpolation.**

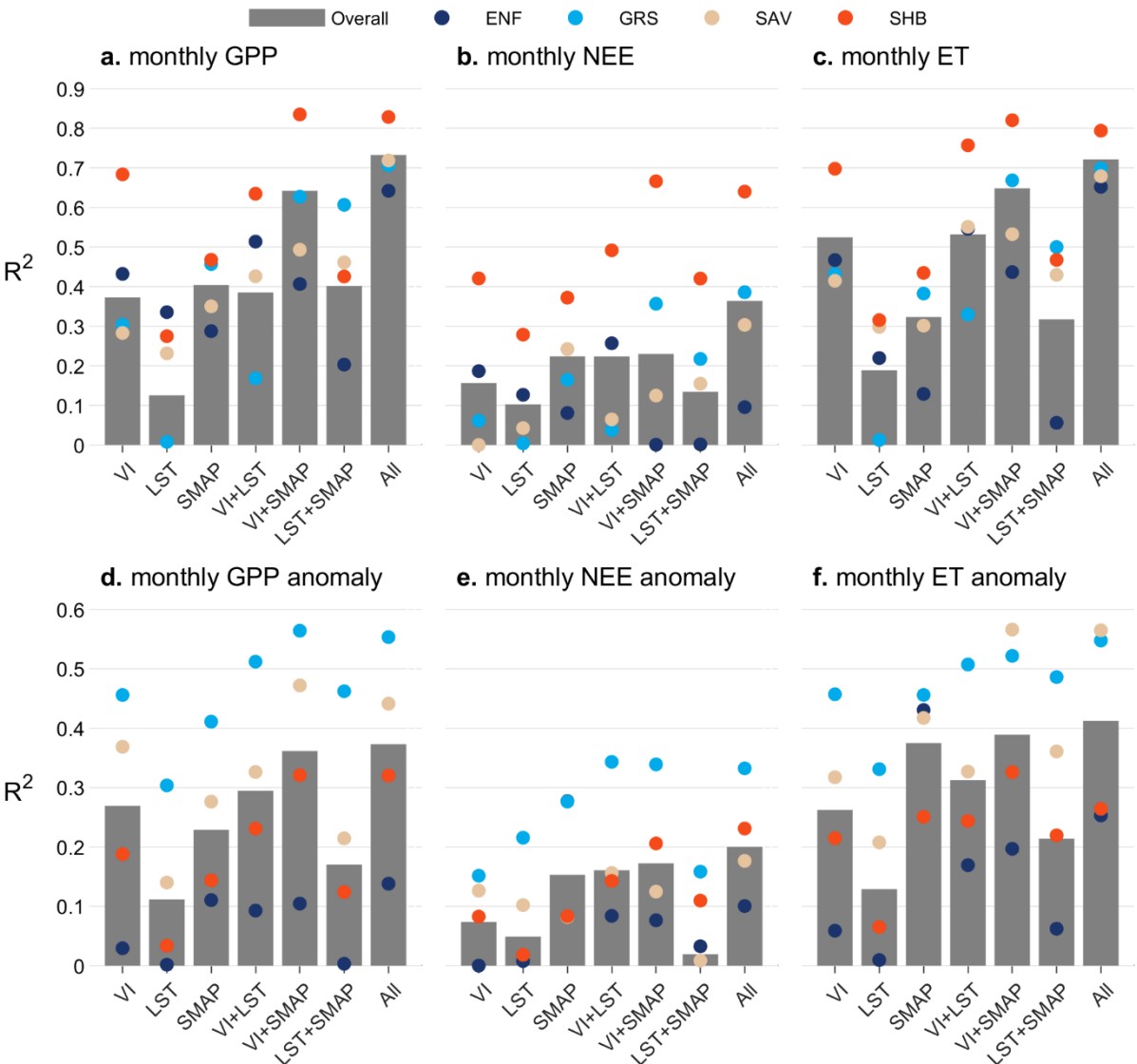

**Figure 8: Coefficient of determination ($R^2$) for models based on different combinations of predictor variables: optical VIs, MODIS LST, and SMAP soil moisture/temperature. Gray bars show overall model $R^2$ (across all sites) while colored dots show model $R^2$ across all sites of a given vegetation type. (a-c) DrylANNd model performance for predicting observed monthly fluxes across all 28 eddy covariance sites. (d-f) DrylANNd model performance for predicting monthly flux anomalies relative to monthly site-means across the 16 eddy covariance sites that cover the full study period (2015-2020). For purposes of calculating monthly site-means, missing monthly data were filled with a spline interpolation after which mean fluxes were calculated for each month during the 2015-2020 study period.**



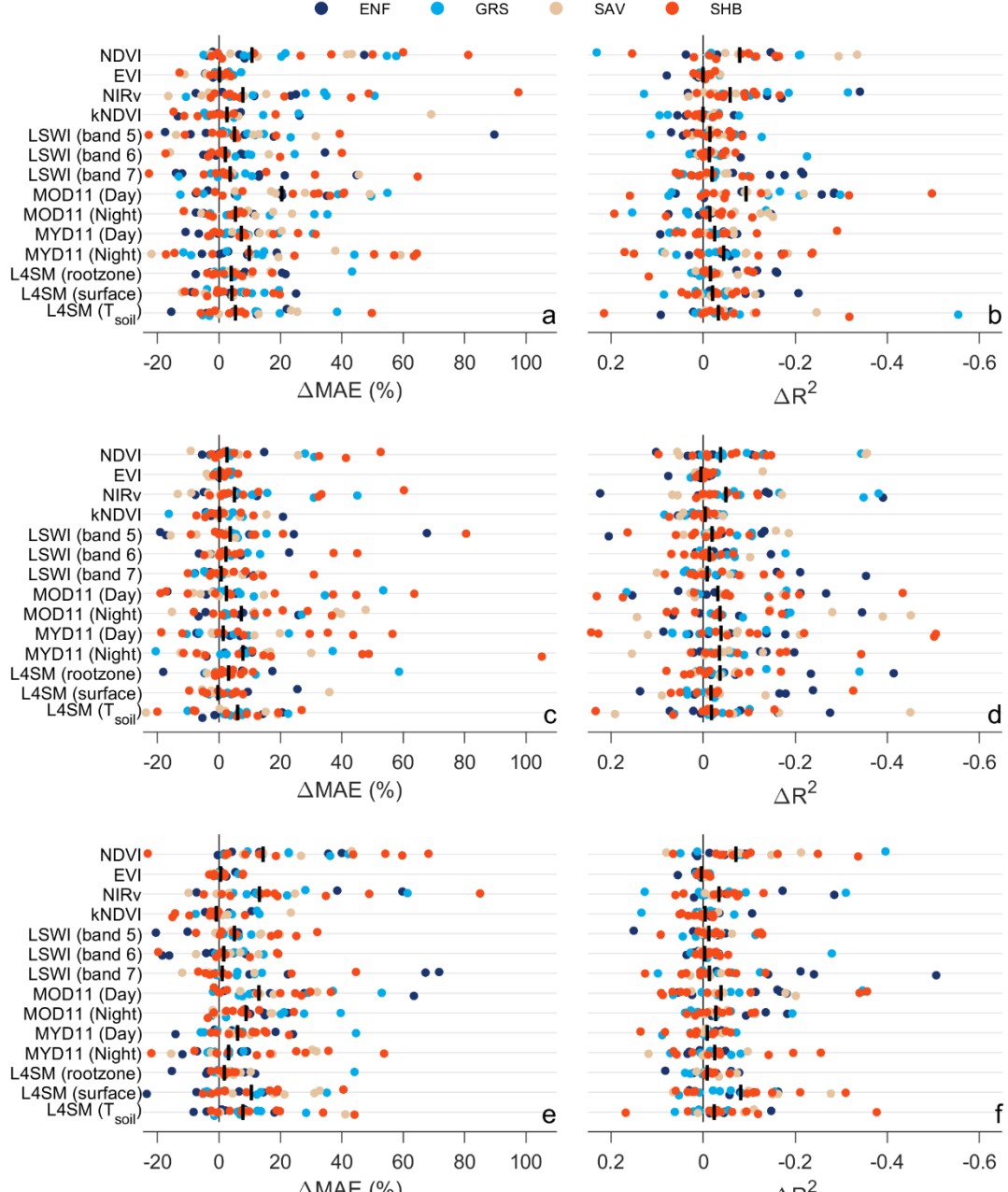

**Figure 9: Site-level (filled circles, colored by vegetation type) and median (black vertical line) change in MAE (ΔMAE) and R² (ΔR²) for DrylANNd model predictions of (a-b) GPP, (c-d) NEE, and (e-f) ET when the information content of each variable is destroyed via random permutation. For each site, 100 random permutations of each variable were performed (each with the same mean and variance as the original variable), model predictions were generated with a randomly-selected ANN (from which that site was excluded from model calibration) from the ensemble for each noise-only permutation, and predictions with the noise-only simulation was compared to those with the original variable.**

Table 1. AmeriFlux (response) and remote sensing (predictor) variables used in the DrylANNd carbon and water flux model.

| Product (site / region) | Variable | Units | Resolution (site / region) |
|---|---|---|---|
| AmeriFlux | Gross primary production (GPP) | g C m$^{-2}$ day$^{-1}$ | - |
|  | Net ecosystem exchange (NEE) | g C m$^{-2}$ day$^{-1}$ | - |
|  | Evapotranspiration | mm day$^{-1}$ | - |
| Optical VIs from MODIS NBAR (MCD43A4 / MCD43C4) | NDVI | - | 500 m / 0.05° |
|  | EVI | - | 500 m / 0.05° |
|  | NIRv | - | 500 m / 0.05° |
|  | kNDVI | - | 500 m / 0.05° |
|  | LSWI (1240 nm) | - | 500 m / 0.05° |
|  | LSWI (1640 nm) | - | 500 m / 0.05° |
|  | LSWI (2130 nm) | - | 500 m / 0.05° |
| LST from MODIS Terra (MOD11A1 / MOD11C1) | Day LST (~10:30 a.m.) | K | 1 km / 0.05° |
|  | Night LST (~10:30 p.m.) | K | 1 km / 0.05° |
| LST from MODIS Aqua (MYD11A1 / MYD11C1) | Day LST (~2:30 p.m.) | K | 1 km / 0.05° |
|  | Night LST (~2:30 a.m.) | K | 1 km / 0.05° |
| SMAP Soil Moisture/Temperature (SMAP L4SM) | Rootzone soil moisture | % | 9 km[a] |
|  | Surface soil moisture | % | 9 km[a] |
|  | Soil temperature | K | 9 km[a] |
| Rangeland Analysis V3 Fractional Cover | Annual grasses & forbs | % | 30 m[b] |
|  | Perennial grasses & forbs | % | 30 m[b] |
|  | Tree | % | 30 m[b] |
|  | Shrub | % | 30 m[b] |
|  | Bare ground | % | 30 m[b] |
|  | Litter | % | 30 m[b] |

[a]Resampled to 0.05° using nearest neighbor for regional scale.

[b]Averaged within 500 m buffer for site-level calibration and within 0.05° grid for regional scaling.
