# Peer review of "Upscaling dryland carbon and water fluxes with artificial neural networks of optical, thermal, and microwave satellite remote sensing"

_Biogeosciences, 2022_

## Author Comment (AC1)

*Dannenberg et al. develop a neural network to predict GPP, ET, and NEE at FLUXNET sites using satellite retrievals of several environmental drivers from several different observing frequencies (optical/thermal from MODIS and microwave from SMAP). DrylANNd is able to predict the GPP and ET seasonal cycle, spatial variability, and, to a lesser degree, their interannual variability. The predictions of NEE are weaker due to satellites not being able to observe respiration. Overall, I find this to be a nice advance and hope this lays the foundation for follow-up studies. The study is very thorough and well-motivated. I support its publication with consideration of points below. With future applications in mind, I encourage the authors to consider several points below as well as some methodological clarifications. Nice work!*
*-Andrew Feldman*

> We thank Dr. Feldman for his supportive and constructive comments. Below, we respond to each comment individually.

*Overall/Major Comments*
*1) What are the desired use cases of DrylANNd? It is a named model, which indicates a future application as the authors briefly mention for a global study in line 117. Machine learning approaches like this require careful calibration and validation, which the authors have done well here. However, if the conditions change to a different region or globally, what needs to change about the inputs as the predictors and predicted variables? Can we rely on the few dryland locations in the Western US to predict other regions when there may be different rainfall seasonality and vegetation types (i.e. African and Australian drylands) or do we need to train the model in each different defined region? Are we restricted to certain datasets to serve as the GPP and ET independent variables?*
*I recommend laying a framework for applications in the discussion by providing more concrete recommendations on how to apply DrylANNd and points about pitfalls that may come about applying DrylANNd at larger spatial scales or other, related to the questions here. I know Section 4.3 may have been an attempt to do this in trying to improve the model overall with new datasets, but I think the authors can expand on that section with regard to these questions and maybe put a more positive outlook on it. Specifically, I recommend being clearer about how DrylANNd can be applied. Next, maybe give a big picture roadmap such as discussing how we may not have as reliable of observation-only data from satellites as we have from FLUXNET to use to train the model on ET or GPP. Therefore, we are constrained to using the model regionally where FLUXNET is available. Perhaps a SIF product (or other) can be used as a predicted variable elsewhere (like Australia) where there are not widespread, publicly available flux tower observations.*

> This is a great suggestion. As Dr. Feldman correctly notes, it would be tough to extrapolate a product trained in one specific dryland region (the western U.S.) to other regions, especially since other regions may have different climatic regimes and since dryland carbon and water fluxes tend to be "unique" to their specific region (Haughton et al., 2018). We think there would be several important use cases of a global dryland product, including: monitoring/forecasting of rangeland productivity (using GPP estimates); improving understanding of the mean and variability of the dryland carbon sink (using the NEE estimates); and monitoring water use, water stress, drought, and fire and/or mortality risk (using the ET estimates).

However, developing a global product would likely require at least two additional steps beyond those that we have taken here: 1) incorporating eddy covariance sites from flux tower networks (e.g., EuroFlux/CarboFlux-Europe, AsiaFlux, and/or OzFlux) that cover other dryland regions for more representative calibration and validation, and 2) using data products that are available globally and that are likely to capture variability in carbon and water fluxes of global drylands (e.g., a land cover product that is not specific to the U.S.).

In the revised manuscript (section 4.3, now retitled "DrylANNd applications and priorities for future dryland model development"), we have added two new paragraphs clarifying both the potential use cases of DrylANNd (paragraph 1) and the necessary steps to accomplish them (the original paragraph from this section plus a new third paragraph addressing needs for global-scale application):

> **New paragraph 1**: "Given the challenges of mitigating and adapting to a changing climate, high-quality remotely sensed carbon and water flux estimates are needed for large-scale monitoring of changes in global ecosystem function and ecosystem services, especially in dryland regions that are warming more rapidly than many other regions (Huang et al., 2017). Ecosystem production estimates provide the means to monitor and forecast rangeland and cropland productivity (e.g., Hartman et al., 2020) and to track changes in the terrestrial carbon cycle (Xiao et al., 2019). Evapotranspiration estimates are needed for monitoring drought and plant water use and water stress (Fisher et al., 2017), which in turn affect both fire risk (Rao et al., 2022) and mortality risk (McDowell et al., 2022)."

> **New paragraph 3**: "Applying the DrylANNd approach at a global scale would require expanding the eddy covariance training sites beyond those used here, which are limited solely to western U.S. AmeriFlux sites. Drylands are generally defined as regions where annual precipitation is insufficient to meet evaporative demand (e.g., P/PET < 0.75), but climates falling within that general definition can have very diverse seasonalities, temperatures, and precipitation regimes (e.g., Fig. 1b). A purely empirical, data-driven model such as DrylANNd would likely struggle to extrapolate to regions beyond those on which it was trained, especially because dryland carbon and water fluxes tend to be more "unique" to their specific region compared to more mesic systems (Haughton et al., 2018). In addition to improving and expanding the input data, a global-scale version of the DrylANNd approach would therefore benefit from expanding the training network to include dryland eddy covariance sites from other global flux networks (e.g., OzFlux, AsiaFlux, and the various European flux networks)."

*2) I want to caution that there may be a drawback in using soil temperature from SMAP L4 as predictor here, especially with regard to the desire to use remote sensing observations to train DrylANNd. The SMAP L4 retrievals are outputs from a land surface model assimilation (see the Reichle et al. 2019 study referenced in the submitted manuscript). While the soil moisture output is highly a function of remote sensing from SMAP's brightness temperatures (especially the 0-*

*5cm product), the soil temperature is likely not as highly influenced by the SMAP observations. Historically, we input soil temperature data from a GMAO model in the process of retrieving L3 SMAP soil moisture – we don't go the other way around to estimate soil temperature. Microwave brightness temperature is a function of physical soil temperature, but more strongly associated with moisture on the surface, and is thus (at least not to my knowledge) not necessarily influencing the soil temperature outputs as heavily in the assimilation. Perhaps the L4 soil temperature output is less of an "effective" remote sensing parameter. I don't think we have good evidence otherwise, though I would be happy for this claim to be refuted which may require a closer look through the literature on assimilating L-band brightness temperature into land surface models. As a consequence, I think SMAP could be overestimated in its ability to explain GPP, NEE, and ET in Figure 8. Since the study's goal is to explain these variables with different observation frequencies from remote sensing instruments, I am not sure the SMAP soil temperature is as appropriate here as the other variables and recommend the MODIS LST or raw infrared data instead.*

This is a very interesting point, and we appreciate Dr. Feldman's insight into the inner workings of the SMAP soil temperature estimates. The main reason that we included soil temperature are: 1) because it plays a very large and important role in soil respiration and thus NEE (Curiel Yuste et al., 2007), including in the dryland environments on which we are specifically focused, 2) because it is also included in the SMAP Level 4 Carbon model as the soil respiration driver (Jones et al., 2017), and 3) because model skill was improved by SMAP soil temperature for all three flux variables. Our choice to use solely remote sensing-based inputs was largely motivated by a desire for products with appropriate spatial and temporal resolutions (~5km and daily or sub-daily) and low latencies. While the soil temperature may be less directly driven by the SMAP microwave signal than soil moisture, since it has the same resolutions and latencies as the soil moisture estimates, we think that the benefit of including it (i.e., its clear biophysical importance) outweighs the potential drawbacks.

In response to this comment, we have taken two steps in the revision:
1) We explicitly acknowledge (Section 2.2, paragraph #3) the point raised by Dr. Feldman (that soil temperature is not as directly related to the microwave signal and is not the main retrieval objective of the L4SM model) but clearly state that we are using it because of its biophysical importance: "While soil moisture (rather than soil temperature) is more directly related to the L-band microwave signal and is the primary retrieval objective for L4SM (Reichle et al., 2017), we also chose to use soil temperature estimates due to the strong dependence of soil respiration (and thus NEE) on soil temperature (Curiel Yuste et al., 2007) and its use in other SMAP-based carbon models (Jones et al., 2017)."
2) We have tested the leverage that the L4SM soil temperature estimates have on DrylANNd model skill and variable importance (compare Fig. 8 to Fig. R1 below). [Note, however, that there will always be some small variation in the skill metrics just because the ANNs have a random element in their initialization and selection of their training and validation sets.] When soil temperature was removed, overall model skill declined for all three flux variables (compare the "All" bars in Fig. R1a-c below to those in the original Fig. 8a-c). Removing soil temperature from the models did not

dramatically alter the skill of the VI+SMAP or LST+SMAP models, though perhaps the skill of the VI+SMAP model tended to be slightly higher when soil temperature was included while LST+SMAP skill tended to be slightly lower when soil temperature was included. The skill of the SMAP-only model, on the other hand, tended to be considerably greater when soil temperature was included. **However, regardless of whether soil temperature was or was not included, the most skillful overall models were, without exception, those that included SMAP data** (the "All" models in Figs. 8 and R1), so our overall conclusions regarding the importance of SMAP for estimating dryland carbon and water fluxes hold with or without the inclusion of soil temperature.

[Figure]

**Fig. R1.** Same as Fig. 8 in the manuscript, but without soil temperature included in the SMAP-based models.

*3) Given that the Western US has seen some unprecedented climatic behavior in the past two decades and especially in the past two years, does this create an issue training DrylANNd on stronger dry response anomalies over 2015-2021? It certainly will be a limitation in applications of predicting future ET and GPP (with regard to my point #1 above).*

We agree with Dr. Feldman that the relatively short calibration period likely imposes some limitations on the ability of DrylANNd, especially given the multi-decade megadrought that much of the western U.S. has experienced so far in the 21[st] century. Essentially, if we were to run predictions forward in the future using our current ensemble of ANNs, we would be assuming a stationary relationship between the remote sensing predictors (VIs, LST, and SMAP) and ecosystem carbon/water fluxes and that that relationship would hold under future climates. Overall, we think this is likely a reasonable assumption for two main reasons. First, during our calibration period (2015-2020), there was actually a reasonably large range of variability in climate with extreme years on both ends of the wet-dry spectrum. In the Southwest, for example (e.g., Fig. R2 below for Arizona), 2019 was one of the wettest years since the mid-1990s (with additional relatively wet intervals from 2015-2016), while 2020 was one of (if not the) driest years on record (Dannenberg et al., 2022; Williams et al., 2022). While it is certainly possible that forthcoming years will be more extreme, the calibration period incorporates a pretty large portion of the climatic variability experienced in the western U.S. Second, because the DrylANNd model ensemble is trained over a large spatial gradient, we get a fairly large sample of the possible "climate space" of the western U.S. If one site experiences climatic conditions outside the range of its training period climate, there is a good chance that another site in the network experienced those climatic conditions. While none of the sites is a perfect analogue of another (e.g., due to differences in community composition, topography, or soil characteristics), this would at least partly mitigate the temporal sampling issue.

We have added a sentence acknowledging the potential limitations of the given these temporal issues (section 4.3, paragraph #2):

"Despite its short calibration and validation period, DrylANNd's training data encompasses much of the climate variability experienced by the western U.S., including both anomalously wet and dry years that may serve as analogues when running the model forward in time as new MODIS and SMAP data are released. However, it is possible that the historically atypical "megadrought" conditions (Williams et al., 2020, 2022; Dannenberg et al., 2022) under which the model was trained may impose limitations on the model's predictive capability."

[Figure]

**Fig. R2.** Percentage of Arizona land area that is drier (D, ranging from mild [0] to extreme [4]) and wetter (W, ranging from mild [0] to extreme [4]) than normal. (source: www.drought.gov/states/arizona).

*4) I think some mention of how spatial scale mismatch between datasets has an influence on results is important. For example, the flux towers have a fetch of <1000m. However, some of the remote sensing products have much larger native resolutions here, which could lead to problems with spatial mismatch of data allowing spatial heterogeneity errors to creep in to the prediction performance estimates. This may be motivation to demonstrate the method entirely with flux tower data at FLUXNET sites and see if similar results occur. I leave that up to the authors to try.*

This is a great point, and one that we thought a lot about during the development process. For the training of the DrylANNd model, we used the finest resolution available for each particular product. For the two MODIS products (the NBAR surface reflectance and the LST), the spatial resolutions come quite close to matching the approximate size of a typical flux tower footprint (500 meter for the NBAR and 1 km for the LST). For the SMAP data (9 km resolution), however, this is clearly not the case. The challenge with testing this using *in situ* data at flux tower sites is that not all flux sites provide soil moisture/temperature estimates, and even fewer provide measurements at multiple depths commensurate with SMAP's definition of the root zone. However, using soil moisture estimates for a subset of towers for which soil moisture was measured at multiple depths (Dannenberg et al., 2022), we compared correlations between SMAP and site-measured soil moisture (both surface and rootzone) and ecosystem carbon and water fluxes (GPP, NEE, and ET) to confirm that they are reasonably similar (Tables R1 and R2 below). [We note that the rootzone definitions are slightly different, 0-100 cm for SMAP and 0-30 cm for site-measured, since even most of these sites do not have probes down to 1-meter depths.] In most cases, correlations were very similar regardless of whether SMAP or site-measured soil moisture were used, and in some cases, SMAP soil moisture was actually *more* strongly correlated with GPP, NEE, and ET than site-measured soil moisture (possibly because the soil moisture estimates are hyper-localized and so the SMAP estimates are actually more representative of the whole area).

That said, we agree that the spatial scale mismatch is an important issue, so we have added a new sentence to section 2.2, paragraph 3 acknowledging this issue: "We note, however, that unlike MODIS resolutions (500 or 1,000 meter), the 9-km SMAP resolution is much coarser than the typical $\sim 1$ km$^2$ (or less) eddy covariance footprint (Chu et al., 2021), so the SMAP soil moisture/temperature estimates used here represent a larger area-integrated average that may not be perfectly representative of conditions inside the flux footprint."

**Table R1**. Correlations of daily site-measured carbon/water fluxes (GPP, NEE, and ET) with daily SMAP rootzone (0-100 cm) and site-measured rootzone (0-30 cm) soil moisture.

| Site | GPP | | NEE | | ET | |
|---|---|---|---|---|---|---|
| | SMAP | Site | SMAP | Site | SMAP | Site |
| US-Mpj | 0.11 | -0.02 | -0.04 | -0.08 | 0.22 | 0.13 |
| US-SRG | 0.31 | -0.04 | -0.25 | 0.05 | 0.32 | 0.11 |
| US-SRM | 0.34 | 0.14 | -0.27 | -0.01 | 0.36 | 0.35 |
| US-Seg | -0.07 | -0.07 | 0.04 | 0.12 | 0.01 | 0.10 |
| US-Ses | -0.11 | -0.18 | 0.09 | 0.23 | 0.06 | 0.16 |
| US-Ton | 0.58 | 0.31 | -0.16 | 0.13 | 0.52 | 0.22 |
| US-Var | 0.70 | 0.46 | -0.37 | -0.04 | 0.68 | 0.36 |
| US-Vcp | 0.09 | 0.14 | -0.31 | -0.51 | 0.17 | 0.08 |
| US-Whs | 0.37 | 0.25 | -0.43 | -0.17 | 0.25 | 0.34 |
| US-Wjs | 0.11 | 0.16 | 0.00 | 0.10 | 0.27 | 0.39 |
| US-Wkg | 0.09 | 0.22 | -0.18 | -0.21 | 0.05 | 0.29 |

**Table R2**. Correlations of daily site-measured carbon/water fluxes (GPP, NEE, and ET) with daily SMAP and site-measured surface ($\sim$0-5 cm) soil moisture.

| Site | GPP | | NEE | | ET | |
|---|---|---|---|---|---|---|
| | SMAP | Site | SMAP | Site | SMAP | Site |
| US-Mpj | -0.05 | -0.04 | 0.21 | 0.06 | 0.39 | 0.25 |
| US-SRG | 0.24 | -0.04 | 0.02 | 0.16 | 0.42 | 0.17 |
| US-SRM | 0.20 | 0.19 | -0.05 | 0.02 | 0.41 | 0.44 |
| US-Seg | -0.15 | -0.07 | 0.25 | 0.19 | 0.20 | 0.25 |
| US-Ses | -0.22 | -0.20 | 0.29 | 0.30 | 0.35 | 0.29 |
| US-Ton | 0.44 | 0.30 | 0.01 | 0.14 | 0.38 | 0.22 |
| US-Var | 0.65 | 0.46 | -0.25 | -0.04 | 0.60 | 0.36 |
| US-Vcp | 0.04 | 0.19 | -0.13 | -0.50 | 0.21 | 0.18 |
| US-Whs | 0.24 | 0.24 | -0.10 | -0.07 | 0.42 | 0.44 |
| US-Wjs | -0.06 | 0.14 | 0.22 | 0.12 | 0.32 | 0.41 |
| US-Wkg | 0.13 | 0.16 | -0.09 | -0.10 | 0.25 | 0.31 |

*Line-specific comments*
*-L30: Wonderfully written introduction*

  Thank you!

*-L65-90: What about effects of biases from soil color contrast and thus soil contamination on the visible signal?*

  Great point! We've added the following sentence to this paragraph (section 1, paragraph #5): "In the open canopies typical of dryland ecosystems, optical VIs are also particularly sensitive to soil background reflectance and the presence of senesced vegetation or standing litter (Huete and Jackson, 1987)."

*-L135: Is the gap filling necessary where the NN approach cannot be used on irregularly sampled data? Such gap filling methods could bias a predictive approach if a functional form is used to gap fill (for example, a look up table that may be based on model assumptions). A noisy insertion could eliminate issues of model assumptions becoming imprinted in the prediction model. Maybe gap filling is not very common in the available time series? What percentage of the different time series are gap filled?*

  Gap filling of the half-hourly flux data is a standard part of the eddy covariance processing pipeline (Papale et al., 2006), and we think it is unavoidable in this case. Most of the data that gets filled in this step are half-hourly periods when there were sudden (but temporary) spikes in the NEE, when turbulence was low (and thus when NEE measurements are unreliable), during precipitation (which interferes with the turbulence measurements), or during brief periods of instrument failure; long, contiguous gaps for which a season-specific look-up table could not be reliably made are not filled by REddyProc. Filling of short gaps is well-validated and based on copious near-term, valid measurements of the flux variables and many strong predictor variables. Since we are aggregating the half-hourly data to daily then monthly scales, the half-hourly gaps would need to be: 1) filled before averaging (as we have done here and as is typical of most flux processing), 2) not filled and then days/months with any missing data excluded, or 3) not filled and then averaged to daily using whatever observations were available during that day. The second option is not feasible because days with filtered observations (due to spikes or low turbulence) are very common and excluding them would leave little remaining data. The third option is also not desirable because often the observations that are filtered out occur at night when turbulence tends to be lower, which would then result in biased estimates of NEE and GPP. We note, however, that the look-up table method does not make *a priori* assumptions about the relationship between NEE and site meteorology (VPD, temperature, radiation). Rather, it fills half-hourly gaps based on available measurements with similar meteorological conditions at that particular site.

*-L173: Note that SMAP products do not retrieve soil temperature, though there are some nuances about the assimilation process in L4. See the major point above.*

We have clarified this point in the revised manuscript (section 2.2, paragraph #3): "While soil moisture (rather than soil temperature) is more directly related to the L-band microwave signal and is the primary retrieval objective for L4SM (Reichle et al., 2017), we chose to also use soil temperature estimates due to the strong dependence of soil respiration (and thus NEE) on soil temperature (Curiel Yuste et al., 2007) and its use in other SMAP-based carbon models (Jones et al., 2017)."

*-L225: what is the "holdout model?"*

We clarified this in the revised text (changes in boldface): "We evaluated model skill based on the coefficient of determination ($R^2$) and mean absolute error (MAE) between model predictions and observations at each site**, using only model predictions generated from the ensemble members in which that site was withheld from training**."

*-L228: Are only 7 data points being used in the interannual timescale prediction? It seems 7 data points from all sites are normalized by taking out their mean and aggregated with other sites to increase sample size (as in Fig. 7). Please clarify in the text.*

Thank you for pointing out this ambiguity. We have updated this paragraph (section 2.4, paragraph #2) to read (changes in boldface): "To assess the ability of the model to capture (inter)annual variability, we calculated **total** annual fluxes **(i.e., the mean daily flux multiplied by the number of days) in** each **of the six study-period** year**s** during the April-October warm season**, with interannual variability defined by comparing the ability of the DrylANNd model to capture variance in flux *anomalies* (i.e., the departure of each year's flux from that site's study-period mean flux) across all sites**."

*-L233: Are these months averaged in all cases for the warm season or is the max used in the case of visible/NDVI like it was for individual months (as stated in line 185)?*

We clarified in the text that we are specifically referring to total annual fluxes in this case, calculated as the mean daily fluxes across the warm season months multiplied by the number of days in the warm season. (See response to previous comment on L228 for the revised text.)

*-L234: A word of caution that SMAP went into safe mode in summer 2019 which led to 1-2 months of loss of data. This is unfortunate because leaving this year in the analysis could bias predictions with biased means. Taking 2019 out removes samples from an already short time series. I encourage the authors to assess the consequences of removing 2019.*

This is a good point, and thanks for reminding us of this issue. As with the SMAP L4C Carbon product, we have decided to retain the soil moisture and temperature estimates during the Safe Mode period since, as you correctly note, removing them would shorten an already short time series, especially for the (inter)annual analysis. We think the issue is likely at least partly mitigated by our monthly time step (since both June and July 2019

would have only partial months of data loss), but we have explicitly acknowledged this issue and its potential impacts on our carbon and water flux estimates in the revised manuscript (section 2.2, paragraph #3): "In 2019, SMAP went into 'Safe Mode' from June 19 through July 23 (Reichle et al., 2022), during which the L4SM model could not assimilate microwave brightness temperature and model estimates would have come solely from the hydrological model forced with meteorological observations. Because this only affects two partial months, we chose to retain soil moisture and temperature estimates during this period, though this may result in slightly higher error or bias in our monthly DrylANNd carbon and water flux estimates for June and July 2019."

*-L246: By the end of the methods, I have not gotten a picture of precisely what the inputs and outputs are. Are the predictors always from remote sensing and the predicted, independent variables are always from FLUXNET? Table 1 does help, but it may help further to add to table 1 that FLUXNET ET, GPP, and NEE are the independent/predicted variables.*

Great suggestion! We have modified Table 1 to include the three variables used from AmeriFlux and to explicitly say in the Table 1 caption which are the response (i.e., AmeriFlux carbon/water fluxes) and which are the predictor variables (i.e., all of the remote sensing products). We have also modified the ANN architecture figure (below) to be less of a general conceptual figure and to more explicitly show our specific structure used here.

[Figure]

*-L293: It might be helpful to mention that time and space are mixed on the left panels of Fig 7 where the spatial patterns might be dominating the good performance there. Only temporal patterns are shown on the right panels.*

> This is a great point. We have modified this section (section 3.3, paragraph #1) to read (changes in boldface): "However, **much of this skill is likely attributable to strong performance at estimating spatial (rather than temporal) variation across the study sites (Fig. 4 and §3.1);** like many remote sensing estimates of GPP and ET (Smith et al., 2019; Biederman et al., 2017; Stocker et al., 2019), DrylANNd struggled to capture the interannual *variability* (i.e., deviations from site mean) of carbon and water fluxes."

*-Fig 8: Can the authors indicate in table 1 or elsewhere which variables are grouped into VI only, LST, and SMAP as corresponding to Fig 8?*

> We have revised Table 1 to make this clearer.

References (excluding those that were already included in the original manuscript):

[revised manuscript text omitted]

---

## Author Comment (AC2)

*Review for "Upscaling dryland carbon and water fluxes with artificial neural networks of optical, thermal, and microwave satellite remote sensing" by Dannenberg et al.*

*Dannenberg et al. present an approach for estimating dryland GPP, NEE, and ET by training an artificial neural network (ANN) with remote sensing signals (optical vegetation indices, thermal observations, and microwave soil moisture/temperature). The study is novel, scientifically sound, well written and within the scope of Biogeosciences. I would recommend this paper for publication but have a few revisions I think should be addressed, mainly around paper presentation and clarification on methodology.*

We thank the reviewer for their helpful and supportive comments, and we are glad that they found our manuscript interesting and useful. We respond to each comment individually below.

*Minor Concerns:*

- *The structure of the introduction and methods have some overlapping material. For example, the fourth paragraph of the introduction in lines 63-72 mentions that plant physiological responses are not necessarily reflected in optical signals, but this paragraph doesn't make the connection between optical VI's that are sensitive to greenness specifically. Discussion of 'greeness'-based metrics failing comes later in the methods section in lines 141-155 but I think it would be useful to draw the connection earlier in the introduction. In addition, the same paragraph in lines 63-72 says "microwave, thermal, and visible wavelengths can capture complementary information about plant and ecosystem stress that is unattainable from optical VIs alone". An explanation as to WHY these indices are useful is available in the methods but could be moved further to the introduction.*

These are excellent suggestions. As suggested, we have moved much of this information from the methods to the suggested places in the Introduction (section 1, paragraph #4), which now reads (new text in **boldface** and moved text in red): "**The normalized difference vegetation index (NDVI), for example,** is the most widely used vegetation index, but it sometimes fails to capture temporal dynamics of carbon and water fluxes in drylands (Yan et al., 2019; Smith et al., 2019; Wang et al., 2022). **While other optical vegetation indices overcome some of the weaknesses of NDVI,** combining different types of remotely sensed observations—such as those from microwave, thermal, and visible wavelengths—can capture complementary information about plant and ecosystem stress that is unattainable from optical VIs alone (Smith et al., 2019; Stavros et al., 2017; Guan et al., 2017). **Land surface temperature from thermal imaging, for example,** is an important determinant of carbon and water fluxes because, among other reasons, both photosynthesis and respiration involve temperature-dependent enzymatic reactions (Farquhar et al., 1980; Atkin and Tjoelker, 2003) and because it is a key indicator of latent heat flux, which cools leaves and land surfaces (Bateni and Entekhabi, 2012)."

- *The final paragraph of the introduction could be rephrased to make the hypothesis/study aim clearer. Specifically, the first sentence states, "Here, we develop and test an*

*approach for data-driven prediction of a full suite of carbon and water fluxes that are specially adapted for drylands using…" but I think this can be much stronger to highlight the value of the study. Something along the lines of, "We aim to improve the prediction of GPP, NEE, and ET based on remotely sensed metrics by using…"*

We like this suggestion and have adopted the language suggested by the reviewer. The first sentence of this paragraph now reads: "Here, we aim to improve estimation of dryland GPP, net ecosystem exchange (NEE), and evapotranspiration using an extensive network of eddy covariance observations and multi-source satellite remote sensing."

• *Somewhere in the methods should include the number of test/train data points used.*

This is a good suggestion, though the answer is a bit complicated. Because each member of the DrylANNd ensemble is trained with one site withheld, and since the period of record varies among the different eddy covariance sites, there is not a fixed number of test/train data points; the exact number will vary depending on which site was withheld from that particular model. For example, the site US-Hn3 has only two years of available records (2017-2018) and thus only 24 monthly flux observations. The 20 ensemble members from which US-Hn3 was withheld would therefore have more available data points in the training and validation sets than would the 20 ensemble members from which the sites with complete records during the study period (e.g., US-SRM, US-Mpj) were withheld. However, in the Methods, we do state the percentages that were used for training (75%) and validation (25%) in the development of each individual ANN; the exact numbers of observations, however, would vary. We now acknowledge this complexity in section 2.3, paragraph #2 (changes in boldface): "Each ANN in the ensemble (§2.4 below) was initiated with randomly assigned weights and biases based on the Nguyen-Widrow method (Nguyen and Widrow, 1990) and with different random subsets of observations for model training (75%) and validation (25%)**, with the precise number of data points used for each individual ANN varying slightly depending on the length of the withheld site's data record**."

• *The final paragraph of the methods discusses the authors approach for testing the importance of predictor variables. Has this approach been used in other studies? Some validation of this approach or references for more information would be useful.*

This is a good point. The methods used to test variable importance in this manuscript are novel but grounded in prior work, such as the stepwise selection approaches that have proved suitable for recognizing the most influential variables in artificial neural networks (Gevrey et al. 2003). To address this in the manuscript, we have made the following change to section 2.4, paragraph #3 (changes in boldface): "Second, we tested the leverage of each time-varying predictor variable by repeatedly (100 times) randomly permuting each variable (thus destroying its information content) and re-running model predictions**, similar to established perturbation and stepwise methods for uncovering the most critical variables in ANNs (Gevrey et al., 2003).**"

- *The color palette of figures could be adjusted to follow more a 'intuitive' color scheme e.g. dark green for ENF – this is not critical but might help with figure readability.*

This is a fair point. We played around with some different color schemes early in the manuscript development process, but we wanted to avoid anything that had a combination of greens and reds to make sure that it's color-blind friendly. Ultimately, we prefer to stick with the existing color scheme since we find it visually appealing and are reasonably confident that the color gradients will be distinguishable by anyone with red-green colorblindness.

*Line edits:*

*Line 37: intensity of water limitation feels like awkward phrasing*

We have changed this to just "water limitation."

*Line 53: It might make more sense to move this like to the end of the last paragraph so someone scanning the paper could easily find "First, Second, Third" in the three paragraphs talking about the unique nature of drylands.*

While we see the reviewer's point, we think that the current placement of the "Several issues…" sentence fits best thematically in its current paragraph.

*Line 54: It might be nice to define mesic*

We have modified this to read (changes in boldface): "…in **wetter,** more mesic systems **where moisture tends to be more plentiful**…"

*Line 59: "the effects of soil moisture stress…" but it's the effects of ALL soil moisture right?*

We have changed this to just say "Soil Moisture…" instead of "The effects of soil moisture stress."

*Lines 53-60: I found this paragraph a little difficult to follow as several sentences are quite long. I think it would be worth revisiting for clarity.*

We have revised this paragraph (including changes made in response to the previous two comments and splitting one of the longer sentences into two shorter sentences).

*Line 67: Satellite-based estimates of fPAR should still be fine, it's just that the plants aren't responding to the increase in light by being more photosynthetically active. I would rephrase this.*

This is a good point. We have removed the part of the sentence that refers to fPAR.

*Line 88: 'however' is unnecessary*

We have made this change in the manuscript.

*Line 90: can be more specific with 'uniqueness'*

We have modified this to read: "… 'uniqueness' of dryland fluxes **to their specific location (i.e., low predictive power of models for sites on which they were not trained)**…" [Our use of the term "unique" in this case is referring to the Haughton et al., 2018 study, but we have tried to be clearer and more explicit here about what we mean by that term.]

*Line 91: 'other places and other types of ecosystems' seems redundant*

We have changed this to just say "…other regions."

*Line 94: 'for example' is unnecessary*

We have made this change.

*Line 97-100: I would rephrase to put the emphasis on the finding of the study, not the author, and just present the citation at the end.*

We have made this change in the revision.

*Lines 113-117: References to sections might be useful*

We have added these throughout this sentence

*Line 117: 'global-scale estimates' – of ecosystem fluxes?*

We have revised this to read "…global-scale carbon and water flux estimates."

*Line 185: 'compositing' is confusing and maybe incorrect?*

We are reasonably confident that we are using this term correctly as maximum value compositing is long-standing technique for aggregating vegetation indices and minimizing noise (e.g., Holben, 1986; Townsend & Justice, 1986). We now include the classic Holben reference directly following "compositing" to make it clear that this is referring to a specific, long-established technique.

*Line 192: this statement deserves a citation*

We have modified this sentence to read (changes in boldface): "ANNs are effective at finding underlying relationships within multidimensional and multisource datasets**, including nonlinear relationships and interactions among predictor variables (Olden et al., 2008)**."

*Line 194: '... predictions of multiple variables.' Deserves a citation*

We have added a reference to Atkinson & Tatnall (1997). Full reference is listed below.

*Line 210: here could be a good place to include the number of test/train data points*

As discussed above in response to Minor Comments, the exact numbers will vary depending on which site was withheld from any given ANN. We therefore think that it makes most sense to just state the percentages of data points that were used in the ANN training and validation sets, though as stated above, we now explicitly state in the revised manuscript that the exact numbers will vary among the ANNs depending on which site was withheld from training.

*Line 328: 'Interestingly' is unnecessary*

Good point! We have removed it.

*Line 333: 'However' is unnecessary*

Deleted.

*Line 340: 'modeling' feels like the wrong term to use here – I think predicting or estimating would be more accurate since modeling implies process based (to me).*

We changed this to "estimating."

*Line 403: 'thermal data' – it might be better to say LST here?*

We have made this change in the revised manuscript.

*Figure 2: I think it would be useful to say what the input variables are in the figure (not just the outputs)*

Great suggestion! We have modified Fig. 2 (below) to be less of a general conceptual figure and to more explicitly show our specific structure and input variables.

[Figure]

*Figure 3: the + indicator is a bit difficult to see/compare with the bars – it might be easier to see in black or a different shape.*

As suggested, we have made the + signs darker (below). We agree with the reviewer that this does indeed make them considerably easier to see.

[Figure]

*Figures 5, 6: I think it would be useful to indicate on the figures somewhere which sites fall under which land cover classification category*

We have now added the three-letter land cover code (ENF, SAV, etc) to the top-right of each subplot to indicate which sites belong to which cover class. (See below for the new version of Fig. 5 with the class labels. The new Figs. 6 and S1-S4 are also now labeled the same way.)

[Figure]

*Figure 7: It's unclear to me what the lines in a and c are*

The lines are showing the linear relationship between predicted and observed GPP and ET for each individual site in the flux network, the idea being to show not just how the model performs at capturing across-site spatial variation but also within-site temporal variation. We have further clarified this in the figure caption (changes in boldface): "**The orange and gray** lines in (a) and (c) show the linear relationship between estimated and observed GPP and ET for each **individual** site **during the six-year training and evaluation period**…"

*Figure 8: Do the lines connecting the scatter points represent anything? If not I would remove*

We had intended the lines to be a helpful visual cue, allowing readers to easily see how the variable importance changed for a particular plant functional type (as well as making

it easier to see where the points overlapped in some cases). However, we can see how this may be either confusing or add extra visual clutter, so we have removed the lines as suggested by the reviewer.

References (excluding those that were already included in the original manuscript):

Atkinson, P. M. and Tatnall, A. R. L.: Introduction neural networks in remote sensing, Int. J. Remote Sens., 18, 699–709, https://doi.org/10.1080/014311697218700, 1997.

Gevrey, M., Dimopoulos, I., and Lek, S.: Review and comparison of methods to study the contribution of variables in artificial neural network models, Ecol. Modell., 160, 249–264, https://doi.org/10.1016/S0304-3800(02)00257-0, 2003.

Holben, B. N.: Characteristics of maximum-value composite images from temporal AVHRR data, Int. J. Remote Sens., 7, 1417–1434, https://doi.org/10.1080/01431168608948945, 1986.

Olden, J. D., Lawler, J. J., and Poff, N. L.: Machine learning methods without tears: a primer for ecologists., Q. Rev. Biol., 83, 171–93, 2008.

Townsend, J. R. G., and Justice, C. O.: Analysis of the dynamics of African vegetation using the normalized difference vegetation index, Int. J. Remote Sens., 7, 1435-1445, 1986.